# UV-Indien Network: Ground-based measurements dedicated to the monitoring of UV radiation over the Western Indian Ocean.

Kevin Lamy[1], Thierry Portafaix[1], Colette Brogniez[2], Kaisa Lakkala[3,4], Mikko R.A. Pitkänen[4], Antti Arola[4], Jean-Baptiste Forestier[1], Vincent Amelie[5], Mohamed Abdoulwahab Toihir[6], and Solofoarisoa Rakotoniaina[7]

[1]LACy, Laboratoire de l'Atmosphère et des Cyclones (UMR 8105 CNRS, Université de La Réunion, Météo-France), Saint-Denis de La Réunion, France
[2]LOA, Laboratoire d'Optique Atmosphérique (Univ. Lille, CNRS - UMR 8518) Lille, France
[3]Space and Earth Observation Centre, Finnish Meteorological Institute, Sodankylä, Finland
[4]Climate Research Programme, Finnish Meteorological Institute, Kuopio, Finland
[5]Seychelles Meteorological Authority, Mahé, Seychelles
[6]Agence Nationale de l'Aviation Civile et de la Météorologie, Moroni, Comores
[7]IOGA, Institute and Observatory of Geophysics of Antananarivo (IOGA) University of Antananarivo, Madagascar

**Correspondence:** K. Lamy (kevin.lamy@univ-reunion.fr)

**Abstract.** Within the framework of the UV-Indien network, 9 ground stations have been equipped with ultraviolet broadband radiometers, 5 of them have also been equipped with an all-sky camera, and the main station in Saint-Denis de la Réunion is also equipped with a spectroradiometer. These stations are spatially distributed to cover a wide range of latitudes, longitudes, altitudes and environmental conditions in 5 countries of the Western Indian Ocean region (Comoros, France, Madagascar, Mauritius and Seychelles), a part of the world where almost no measurements have been made so far. The distribution of the stations is based on the scientific interest of studying ultraviolet radiation, not only in relation to atmospheric processes, but also in order to provide data relevant to fields such as biology, health (prevention of skin cancer), and, agriculture. The main scientific objectives of this network are to study the annual and inter-annual variability of the ultraviolet (UV) radiation in this area, to validate the output of numerical models and satellite estimates of ground-based UV measurements, and to monitor UV radiation in the context of climate change and projected ozone depletion in this region. A calibration procedure including three types of calibrations responding to the various constraints of sustaining the network has been put in place and a data processing chain has been set up to control the quality and the format of the files sent to the various data centres. A method of clear-sky filtering of the data is also applied. Here, we present an intercomparison with other datasets as well as several daily or monthly representations of the UVI and cloud fraction data to discuss the quality of the data and their range of values for the older stations. (Antananarivo, Anse Quitor, Mahé and Saint-Denis). Ground-based measurements of UV index (UVI) are used to validate satellite estimates (Ozone Monitoring Instrument (OMI), the TROPOspheric Monitoring Instrument (TROPOMI), the Global Ozone Monitoring Experiment (GOME) and model forecasts of UVI (Tropospheric Emission Monitoring Internet Service, TEMIS and Copernicus Atmospheric Monitoring Service, CAMS). The median relative differences between satellite or model estimates and ground-based measurements of clear-sky UVI range between – 34.5 % and 15.8 %. Under clear skies, the smallest UVI median difference between the satellite or model estimates and the measurements made by ground-based

instruments is found to be 0.02 (TROPOMI), 0.04 (OMI), -0.1 (CAMS) and -0.4 (CAMS) at St-Denis, Antananarivo, Anse Quitor and Mahé, respectively. The diurnal variability of UVI and cloud fraction, as well as the monthly variability of UVI is evaluated to ensure the quality of the dataset. The data used in this study is available at: https://doi.org/10.5281/zenodo.4811488 (Lamy and Portafaix, 2021).

## 1 Introduction

The sun produces a broad spectrum of radiation, in which wavelengths between 100 and 400 nm correspond to ultraviolet radiation (UVR). The UVR that reaches the Earth's surface has a direct effect on human health, terrestrial and aquatic ecosystems, and the degradation of materials (Bernhard et al., 2020).

UV has important effects on human health. These effects can be beneficial or harmful. For example, human exposure to UV-B ensures the synthesis of vitamin D, which is essential for bone mineralization. UVR is also known to be a powerful virus killer, which can explain some epidemiological pattern (Duan et al., 2003). On the other hand, UV-B can have a harmful effect by altering the DNA of cells. For humans, an increase in UV-B radiation increases the risk of skin cancer and the occurrence of cataracts and weakens the immune system.

The UV Index (UVI) was defined by the World Health Organization (WHO, 1995) to measure the amount of sunburn-producing UV radiation that reaches the surface. If the UVI is above 3, protection is necessary and, above 8, prolonged exposure can become dangerous. Indices above 11 are considered extreme.

Naturally, the geographical area corresponding to the tropics of the southern hemisphere (tropical and ocean-dominated environment) exposes populations to high levels of UV radiation. This is because the zenith angles of the sun close to noon are very small all year round and the amounts of stratospheric ozone or atmospheric aerosols are lower than at higher latitudes in continental regions. This is the case in most of the southern Indian Ocean (La Réunion, Madagascar, Mauritius, Seychelles, Comoros). UV indices in these regions can exceed 10 almost all year round, which implies serious consequences for the health of the populations. For example, dermatologists in Reunion Island (21°S, 55.5 °E) have noted a rapid increase in sun-induced skin lesions (Observatoire Régional de la Santé de La Réunion, 2008). The number of these lesions is currently tripling every 10 years, with an accelerating progression, and more than 2% of the population is affected. Radiation doses received by particularly exposed populations (children and outdoor workers) exceed dangerous thresholds in this region of the Indian Ocean almost all year round (Wright et al., 2013). Finally, it can be noted that all skin phototypes are concerned, including photo-types 5 and 6 (Sitek et al., 2016). An important part of this observed increase can be explained by lifestyle changes, with an expansion of seaside activities and a constant increase in exposure to the sun. The development of tourism, especially local tourism, is a determining factor.

When UV radiation penetrates the atmosphere, its intensity along the atmospheric path and on the ground depends on various parameters, including geometric or geographical factors such as the solar zenith angle (SZA), latitude and altitude. Cloud cover plays an important role, generally reducing the intensity of radiation, although under certain conditions cloud cover can also increase the intensity, as it is the case of fractional cloud cover (Sabburg and Wong, 2000). Aerosols, small particles of natural

(marine aerosols, volcanic) or anthropogenic (biomass fires, fossil fuel combustion) origin, scatter and absorb UV radiation. If present in large quantities, they can strongly affect surface UV radiation or radiative forcing. The reflectivity of a surface, or albedo, is also involved in modulating the intensity of radiation and, finally, the concentrations of certain gases, such as ozone, in the atmosphere play an essential role. The amount of ozone ($O_3$) in the stratosphere largely drives UV levels on the ground because ozone has a high absorption capacity in the UV wavelengths. $O_3$ is produced in the tropics under the influence of strong solar radiation and is transported towards the poles by the Brewer-Dobson circulation (BDC).

Ozone concentrations have been significantly depleted over the past 35 years due to emissions of halocarbons, which are both powerful greenhouse gases (GHGs) and ozone-depleting substances (ODS). The Montreal Protocol (1987) and its amendments are an undisputed success of environmental policy making and have slowly brought ODS amounts down almost to historical levels (Chipperfield et al., 2015).

However, the evolution of stratospheric ozone beyond the middle of the 21st century, when ODS emissions will be minimal and their atmospheric concentrations will continue to decrease, will be largely determined by the concentrations of certain greenhouse gases (GHGs) such as $CO_2$, $N_2O$ and $CH_4$ (Eyring et al., 2013). On the other hand, the increase in $CO_2$ is expected to accelerate the BDC (Butchart, 2014; WMO, 2018). Ozone will then be transported more rapidly from the tropics to the poles and thus the total ozone column (TOC) will increase in mid-latitudes and decrease in the tropics. Since UV radiation at the surface is directly dependent on the total ozone column, UV can be expected to decrease in mid-latitudes and increase in the tropics (Butler et al., 2016; Lamy et al., 2019). Therefore, climate change, linked to the accumulation of greenhouse gases in the atmosphere, is also one of the phenomena identified to explain the increasing levels of exposure to UV radiation in the tropics.

The western Indian Ocean is mainly located in the tropical region of the southern hemisphere. This part of the world is therefore potentially impacted by the changes in radiation described above. However, this geographical area suffers from a very large measurement deficit, both for ozone and UV radiation on the ground. Studies associated with these topics are sparse in the literature. The originality of Reunion Island is that it is located in the heart of this region and and has benefited from high-quality measurements of ozone and aerosols for several decades thanks to OPAR (Reunion's Atmospheric Physics Observatory). Within the framework of various programmes funded by Europe, Reunion Council or CNES (French Space Agency), the University of Réunion Island has developed a specialized network for the measurement of UV radiation and cloud cover in the south-western part of the Indian Ocean, known as the UV-Indien network. The data-sets presented here are original and, as mentioned, concern an extremely poorly measured region of the globe, which highlights their importance. They include the UV index at 9 stations, the cloud fraction measured at 4 stations. These measurements are ideally suited for assessing the effect of UV radiation on the health of people or ecosystems in this region.

## 2 UV-Indien Network

### 2.1 Scientific Objectives

The UV-Indien measurement network proposes to answer the following scientific questions:

- What is the annual and inter-annual variability of UV radiation at the scale of the Western Indian Ocean area?

- How is UV radiation impacted by regular episodes of ozone-rich or ozone-poor air mass filaments, over different places?

- Are chemistry-transport models or regional climate models, coupled with radiative transfer models, reliable for predicting the evolution of UV radiation in the southern tropics?

- How will UVR evolve by the end of this century in the Indian Ocean, in the context of climate change and the predicted 3-4% decrease in ozone ?

## 2.2 Selection and Characterisation of Sites

Although the selection of the instrumented sites of the UV-Indien network was conditioned by financial, technical and logistical feasibility constraints, the choice of the sites where the stations were implanted was also based on several scientific criteria: a spatial distribution of stations that is as homogeneous as possible over a region where the majority of the land area is located in the western part (an essential criterion for obtaining the highest geographical representativeness, in order to be able to validate the outputs of climate models for UV forecasting in the region); a diversity of environmental conditions (between urban and rural sites, different degrees of pollution, close to the coast or not, at low or high altitude); the presence of scientific partners on site; and the interest of UV measurements for other scientific disciplines. For example, the site of Anse Quitor, Rodrigues is situated on the roof of the François Legat turtle reserve. This reserve aims to preserve giant tortoises and monitor their behaviour and biology, so it is of interest to use environmental data such as UV radiation and cloudiness. The sites of the UV-Indien network are presented in Table 1 and are indicated on the map (Figure 1).

The site of Saint-Denis, Réunion, is the main site of the UV-Indien network. UV measurements have been carried out since January 2009 with the Bentham DTMc300 spectroradiometer (affiliated with the Network for the Detection of Atmospheric Composition Change (NDACC)) and since December 2016 for the Kipp & Zonen UVS-E-T radiometer. Cloudiness measurements have been carried out since October 2016. This site has the advantage of being co-located with several other instruments allowing detailed studies of physical processes. These include the UV-Visible spectrometer SAOZ (Système D'Analyse par Observations Zénithales) (Pastel et al., 2014; Toihir et al., 2018), which measures the total column of ozone and nitrogen dioxide and is part of the NDACC network, or the sun-photometer CIMEL (Kabanov et al., 2001), which carries out measurements of the optical and microphysical properties of aerosols and is part of the AERONET network (AErosol RObotic NETwork, Holben et al. (1998)). The environment of Saint-Denis is urban and slightly polluted by heavy road traffic despite the regular advection of air by the trade winds. This site is located on the coast, at an altitude of 82 m asl and is surrounded by high reliefs of 500 to 1500 m altitude. The climate is tropical and marked by two distinct seasons: a rainy season in the southern summer and a dry season in the southern winter dominated by strong trade winds . The climate of all sites generally follows this pattern, with some exceptions mentioned in the following paragraphs.

The Maido site, Reunion Island, is the site of the Maido Atmospheric Observatory (Baray et al., 2013). This site has been instrumented to perform UV measurements since February 2020 with a Kipp&Zonen SUV-E radiometer. This site has the advantage of being highly multi-instrumented; in-situ measurements of the composition of the atmosphere are carried out ($CO_2$,

CO, CH4, ...), meteorological measurements, measurements of atmospheric profiles of temperature, water vapour or ozone (via LIDAR or radiosoundings). This site is located at 2200m asl, outside the boundary layer, the environment is mountainous and uninhabited, the climate is generally dry. The vegetation is sparse and consists of lichens and small shrubs, the soil is composed of lapilis (Strasberg et al., 2005). This area is classified as shrubland on lapillis. This area has often affected by fire episodes.

The site at Anse Quitor, Rodrigues, has been operational since June 2017 for UV measurements using a Kipp&Zonen
UVS-E-T radiometer, and since September 2019 for cloudiness measurements.This site is located at 32m asl near the coast, the climate is very dry, rainfall is low for most of the year, vegetation is sparse and the terrain is slightly mountainous. The Mahé, Seychelles site has been operational since November 2017 for UV measurements using a Kipp&Zonen UVS-E-T series radiometer. Cloudiness measurements are not yet possible, as the deployment of the network has been delayed by the COVID-19 health crisis. The instrumented site is located at 15m asl, near the coast. The instrument is positioned at Victoria International
Airport, in an urban area. The island of Mahé has an equatorial climate all year round, as it is located very close to the equator. The vegetation of the instrumented area is sparse and consists of grasses and small shrubs and can be classified as savanna (Hume et al., 2013).

The site in Antananarivo, Madagascar has been operational since December 2016 for UV measurements using a Kipp&Zonen UVS-E-T series radiometer. Cloudiness measurements are available since April 2019. The instrumented site is located at 1370m
asl in the highlands of Madagascar. Located in a densely populated urban area, the air quality of the site can be very degraded, frequent pollution peaks are observed as well as the presence of smog (Gorremans and Masquelier, 2018). The radiometer and camera are installed on the roof of the Geophysical Institute and Observatory of Antananarivo (IOGA).

The instrumental site of Diego Suarez, Madagascar has been equipped with a Kipp&Zonen SUV-E radiometer since November 2019, the deployment of the camera there has been delayed due to technical incidents following the passage of tropical
cyclone Belna and due to the recent health crisis. The site is located at an altitude of 35m asl, close to the coast, in a semi-urban area, where air pollution is generally low.

The site of Fort Dauphin, Madagascar is equipped with a Kipp&Zonen SUV-E radiometer and a SkyCamVision camera since January 2020. This site is very similar to the one in Diego-Suarez, it is located at an altitude of 10m asl, close to the coast, in a semi-urban area, where air pollution is generally low. However, there is a 10 degree latitude difference between this site
and Diego-Suarez.

The site of Moroni, Comoros, is equipped with an SUV-E radiometer and a SkyCamVision camera since December 2019. The instruments are located on the roof of the Agence Nationale de l'Aviation Civile et de la Météorologie des Comores and are therefore co-located with meteorological records of rainfall and temperature. The site should soon be equipped with a SAOZ that will allow the monitoring of the total ozone column. The site is located at an altitude of 12 m asl, in an urban area and
close to the coast, where the air is generally not very polluted.

Juan de Nova Island is located in the centre of the Mozambique Channel is an integral environmental reserve and its access is strictly regulated. There are no inhabitants on this island. This site is equipped with a Kipp&Zonen radiometer since April, 2019. The purchase of the equipment and its installation was supported by the French space agency (CNES), which is very interested in this type of data for the validation of satellite sounders. Access to this island is difficult, data are stored there but

can only be retrieved locally, so they are brought back at least once a year. However, this was not the case during the recent health crisis. This completely flat coral desert island has an area of 4.80 km$^2$ with very sparse vegetation. The instrumented site is at an altitude of 10 m asl.

## 3    Instrumentation and Data

### 3.1    UV-Indien Instrumentation

Each station in the UV-Indien network is equipped with at least one broadband UV radiometer. This is usually supplemented by a all-sky camera. The radiometers installed at the UV-Indian network sites are the Kipp&Zonen UVS-E-T or the Kipp&Zonen SUV-E. The camera installed is a SkyCam Vision developed by Réuniwatts. Some stations of the UV-Indian network are on the same site as other measurement networks (NDACC, ICOS, AERONET, etc.) and therefore benefit from more complementary measurements. The specificities of the instruments of the UV-Indien network, only, will be detailed here; namely the

radiometers and the camera.

### 3.1.1    Instrument Details

The Kipp Zonen UVS-E-T radiometer is a radiometer designed to have a spectral response close to the erythemal action spectrum (ISO 17166:1999 / CIE S 007/E-1998). The detection system consists of an optical filter and a phosphor, which determines the spectral response. The phosphor, sensitive to UV light, emits light which is then detected by a photodiode. The

resulting signal is amplified and the output voltage, multiplied by 40m$^2$/W, gives an estimate of the UVI. The angular response of the radiometer is a function of the cosine of the zenith angle. According to the manufacturer: the expected daily uncertainty is less than 5%, the measurement drift of the sensor is less than 5% per year, the directional response error is less than 2.5% and the non-linear error is less than 1%. The Kipp&Zonen SUV-E is the new version of the UVS-E-T from the same manufacturer. This new radiometer no longer has a desiccant that needs to be changed regularly and has a digital interface. It is also designed

to have a spectral response close to the erythemal action spectrum (ISO 17166:1999 / CIE S 007/E-1998). Its detection system is similar to that of the Kipp&Zonen UVS-E-T. The expected daily uncertainty, according to the manufacturer, is less than 5%. The measurement drift of the sensor is less than 5 % per year, its directional response error is less than 5 W/m$^2$ and its non-linear error is less than 1%. The broadband radiometers are calibrated about every 2 years (depending on logistical and current health conditions) according to a procedure that is specified in the Section 3.1.2. The SkyCam Vision camera built

by Réuniwatts is an all-sky camera that acquires hemispheric images of the sky in the visible spectrum every minute. The CMOS acquisition sensor has a resolution of 2048 x 2048 pixels and the acquisition frequency is usually 30 seconds but can be modified. The cloud fraction calculated by the camera is obtained according to the ELIFAN algorithm, developed at CNRS, France (Lothon et al., 2019). The algorithm is based on the application of criteria and thresholds on the distribution of R/B ratios, and the R/B ratio of each pixel. Two treatments are performed: an "absolute" treatment, and a "differential" treatment

with respect to a library of clear sky images built up initially. The steps of the process are as follows:

- Limitation of the processed image contour

- Application of masks: solar and object

- Detection of clear sky and fully cloudy images

- Discrimination between clear sky and fully cloudy images through absolute or differential threshold process

More information on this camera is available on the manufacturer's website: http://www.reuniwatt.com/ and in Lothon et al. (2019) for the ELIFAN algorithm. The Bentham DTMc300 spectrometer consists of a dual monochromator giving irradiance measurements with a wavelength step of 0.5 nm and a full width at half maximum of 0.5 nm. The measurements are cosine corrected, temperature stabilised and calibrated every three months with standard 1000 W lamps that can be traced back to the National Institute of Standards and Technology. This instrument also underwent intercomparison during a QASUME (Quality

Assurance of Ultraviolet Measurements in Europe) campaign in 2013. The Bentham used in the UV-Indien network, located in St-Denis de la Réunion, Moufia site, showed a bias of about 3% (Bentham measurements lower than measurements with the reference instrument). The uncertainty on irradiance is about 5% for a coverage factor of k=2 (Brogniez et al., 2016).

### 3.1.2   Maintenance and Calibration Protocols

The radiometers do not require any particular maintenance, except a regular cleaning of the dome. For older models of the

Kipp and Zonen radiometer (Kipp & Zonen UVS-E-T), it is also necessary to change the desiccant approximately every 6 months or when necessary (colour change). This is done by our local partner. In general, it is usual to recalibrate this type of radiometer every 2 years to avoid drift. The calibration protocol is identical for the UVS-E and SUV-E radiometers. However this protocol depends on the type of radiometer and the type of calibration most recently performed for each radiometer. Table 3 shows the last calibration performed for each radiometer. There are three types of calibration: the manufacturer's calibration,

the PMOD/WRC calibration named Davos calibration hereafter, and the Bentham calibration. The manufacturer's calibration is used from the start of the instrument operation and for a period of approximately 2 years. This period can be extended to 3 years when logistical constraints do not allow the recovery of the radiometer at its installation site. It should be noted that many of the stations in the network are costly to transport and are located in countries and regions that are difficult to access, with little or no freight service. Even though this calibration step is essential for the quality of the measurements, we have

to adapt our constraints. The manufacturer's calibration is a function of the solar zenith angle and the total ozone column. This calibration is performed with a Xenon lamp, a monochromator (Oriel Cornerstone MS257) and a sensor consisting of a calibrated silicon photodiode. The calibration coefficient is obtained following several monochromatic measurements from 280 nm to 400 nm with an increment of 1 nm. In order to take both the actual response of the instrument and the ideal response into account, a radiative transfer model is used to estimate the sensitivity as a function of several atmospheric spectra corresponding

to zenith solar angles ranging from 0 to 70° in 5 degree steps and from 260 to 400 DU in 10 Dobson Unit steps. When the manufacturer's calibration expires, the instrument is re-calibrated in two different ways. If our budget can cover the cost, the instrument is sent to the World Radiation Centre in Davos, Switzerland (PMOD). Otherwise, it is recalibrated directly on the

Moufia station in Reunion Island by direct comparison with the calibrated values of the Bentham spectro-radiometer. The calibration performed at the PMOD/WRC is carried out in 3 steps, allowing the weighted irradiance to be calculated according to the following formula:

$$E_{\text{CIE}} = (U - U_{offset} \cdot C \cdot f_n(\text{SZA}, \text{TOZ}) \cdot \text{coscor}$$

U is the value of the raw voltage signal of the radiometer. The calibration therefore allows the following 4 parameters to be determined:

– C, the absolute calibration factor determined for a solar zenith angle, SZA, of 40° and total column ozone (TO3) of 300 DU.

– The conversion function (normalised), fn, which converts from the detector weighted solar irradiance to erythemal weighted irradiance.

– Coscor, a factor that corrects the detector cosine error.

– The dark offset, U offset, which is determined every night.

More details are available at: https://www.pmodwrc.ch/en/world-radiation-center-2/wcc-uv/uv-radiometer-calibration-services/

The Bentham calibration consists of co-locating the radiometers with the Bentham DTMc300 at the St-Denis site for a period of 3 to 4 months. The relative differences between the radiometers and the Bentham measurements are then calculated, and are also classified by SZA band (about $\pm 5°$). The average of these relative difference bins allows us to obtain a calibration coefficient depending on the SZA. The average of all relative differences (without distinction of SZA) is also calculated to obtain an average calibration coefficient. This calibration could be refined by also classifying the biases according to the total ozone column observed during the measurement but this has not been done. In order to obtain sufficient measurement points covering all observed total ozone column values in the Indian Ocean region, the intercomparison period would need to last at least one year to capture the annual ozone cycle. A period of one year is too long for the capabilities and needs of the Indian UV network and therefore the Bentham calibration does not include the total ozone column discrimination.

A Bentham calibration was performed for the UVS-E-T 15-0124 and SUV-E-1800 (27/28/29/30) radiometers between June 2019 and Decembrt 2019 and average relative differences ranging from about -10% to +5% were obtained before recalibration. The SUV-E radiometers (previously calibrated by the manufacturer) tend to underestimate the UVI (by about 10%) with respect to the BENTHAM measurements. The UVS-E-T 15-0124 radiometer (recalibrated DAVOS-PMOD/WRC) tends to overestimate UVI by about 3% compared to the Bentham measurements. The relative differences depend on the SZA and tend to become more pronounced as the SZA increases. After recalibration of the UVS-E-T and SUV-E radiometers using the calibration coefficients described above, a second comparison with the BENTHAM showed average relative differences of $\pm 4\%$. The cameras do not require any particular maintenance except for regular cleaning of the window.

### 3.1.3 Data Processing

With the exception of the station on the island of Juan de Nova, raw data are transmitted daily. These data then undergo a first processing stage in which a first quality control eliminates acquisition errors or outliers. The data files are then homogenised in order to be recalibrated. Depending on the most recent calibration of the instrument, the data are recalibrated according to the manufacturer's calibration procedure, DAVOS or Bentham. They are then reformatted in the format requested by the various distribution platforms (WOUDC or Zenodo). The frequency of updating of the online databases is 3 to 6 months. For the Juan de Nova station, an annual rotation is planned in order to physically recover the data and carry out the maintenance of the instrument. The radiometer data are filtered to define the UVI measurements made in clear sky. This filtering is done manually for 1h intervals. The filtering process is as follows: each daily UVI and cloud fraction profile is plotted together with an estimate of the clear sky profile using the Madronich analytical formula (Madronich, 2007). To calculate the Madronich UVI estimate, we use the total ozone column from either the OMTO3d satellite product or a co-located ground-based instrument (the SAOZ for the St-Denis station). An observer then selects the one-hour windows considered as clear sky according to the following criteria:

- Difference between observed and estimated UVI (according to the Madronich Analytical formula) higher than 10%

- Presence of clouds, a cloud fraction threshold of 0.25 is set.

- Shape and regularity of the UVI curve during the day. A Gaussian (or semi-Gaussian) curve indicates a day (or half-day) not affected by the presence of clouds. As clouds generally have a high temporal variability, rapid development of clouds usually results in rapid variations of UVI over a few minutes.

- Cloud cover can also be quasi-homogeneous and quasi-constant over the day, which does not lead to sudden variations in UVI. This case is also excluded from the filtering.

Figure 2 represents the time period covered by all the instruments of the UV-Indien network.

## 4 Data Overview

### 4.1 Comparison Against Existing UVI Product

#### 4.1.1 Other UVI estimates available in the Indian Ocean region

UVI ground-based observations offer high temporal and spectral resolution and precision. However, to study the impact of climate change on the UVI, global estimates of UVI are also needed. Satellites and Global Forecast Models provide such estimates.

To compute the UVI at the surface, the Global Forecast Model uses a radiative transfer model (RTM) or a look-up table generated by an RTM. Multiple parameters are required, such as TOC, aerosols, CF and SZA. To determine the UVI at the

surface, satellite UV estimates are calculated using a combination of RTM calculations and measurements. In order to compare ground UVI measurements with the UVI product from satellites or forecast modelling, we have gathered together the multiple datasets presented in Table 2.

The Bentham spectrometer (called UVI-BENTHAM hereafter) and Kipp&Zonen Radiometer (called UVI-RADIO hereafter) constitute the ground-based instruments part of the UV- network. These instruments will be compared with satellite surface UV products (TROPOMI, OMUVBG and GOME-2) and forecast model products (CAMS and TEMIS).

The TROPOMI instrument is onboard the Sentinel-5 Precursor (S5P) polar-orbiting satellite launched on 13 October 2017 as part of the EU Copernicus programme. TROPOMI surface UV radiation products include irradiances with daily integrals at four different wavelengths, and dose rates with daily doses for erythema (CIE Standard, 1998) and vitamin D synthesis (Bouillon et al., 2006) action spectra. All parameters are calculated for overpass time, solar noon time, and for clear-sky conditions (no clouds, no aerosols). The TROPOMI UV algorithm (Lindfors et al., 2018) is based on two pre-computed look-up tables (LUTs): The first is used to retrieve the cloud optical depth from the measured 354 nm reflectance. This cloud optical depth, the total ozone column from the TROPOMI level 2 total ozone column product (Garane et al., 2019), the surface pressure, the surface albedo and the SZA are then used as input to the second LUT from which the UV irradiances and dose rates are retrieved. A post-correction for the effect of absorbing aerosols (Arola et al., 2009) is applied to the irradiances. The ground resolution for the UV products is 7.2 x 3.5 km$^2$ (5.6 x 3.5 km$^2$ since 6 August 2019) at nadir. Only the estimates corresponding to the time of the overpass are chosen here. The UVI estimated in clear-sky conditions and in all-sky conditions will be used in this study and will both be called UVI-TROPOMI for ease of reading. UVI-TROPOMI computed for clear-sky conditions will be compared against UVI-RADIO or UVI-BENTHAM measured in clear-sky conditions and UVI-TROPOMI computed for all-sky conditions will be compared against UVI-RADIO or UVI-BENTHAM measured in clear-sky conditions.

OMUVBG is a product derived from the Ozone Monitoring Instrument (Levelt et al. (2006), Tanskanen et al. (2007), Arola et al. (2009)). It is based on the Total Ozone Mapping Spectrometer (TOMS) algorithm to retrieve TOC. TOC, measured with OMI, along with climatological albedo, ozone and temperature profile, elevation, and SZA are used as input to an RTM to compute a first estimate of the UVI under clear skies. The measured reflectance at 360 nm made by the OMI is used to estimate a cloud modification factor (CMF). The CMF represents the attenuation of UV radiation by clouds and non-absorbing aerosols. The clear-sky UVI computed previously is multiplied by the CMF to obtain an estimation of UVI in all-sky conditions (called UVI-OMUVBG hereafter). This estimate corresponds to the satellite overpass time. Absorbing aerosols are only corrected if the aerosol index is higher than 0.5 and if the measured reflectance at 360 nm is lower than 0.15 (Tanskanen et al., 2006). This could lead to overestimation for regions affected by absorbing aerosols (Tanskanen et al., 2007). UVI-OMUVBG computed for clear-sky conditions will be compared against UVI-RADIO or UVI-BENTHAM measured in clear-sky conditions and UVI-OMUVBG computed for all-sky conditions will be compared against UVI-RADIO or UVI-BENTHAM measured in clear-sky conditions.

The offline surface UV is a product derived from the measurements of the GOME-2 instruments on-board the METOP-B and METOP-C satellites. The offline surface UV contains multiple variables related to UV radiation and human health: UVI, integrated UVB and UVA, and daily doses derived from different biological weighting functions (erythema, DNA damage,

plant damage and vitamin-D synthesis). TOC and cloud measurements provided by the AC SAF Total Ozone product and the Advanced Very High Resolution Radiometer (AVHRR-3) reflectances are used with an RTM to compute the offline surface UV product (Kujanpää and Kalakoski, 2015). The product is given on a regular 0.5 x 0.5° grid and UVI is computed at local solar noon (called UVI-GOME hereafter).

The Integrated Forecasting System (IFS) of the Copernicus Atmosphere Monitoring Service (CAMS) has been providing UV forecasts since 2012 (Bozzo A., 2015). More precisely, it provides the spectral UV with a spectral resolution of 5 nm and the UVI by integrating the spectral UV according to the erythema action spectrum (CIE Standard, 1998). The UV irradiances and UVI are produced in clear-sky and in all-sky conditions (hereafter called UVI-CAMS). For UVI-CAMS, new forecasts are available every 12 hours. The model output has a time step of 3 hours. From the first forecast of the day, which starts and is initialized at 00:00:00, we take the first, second, third and fourth time steps of the model (00:00:00, 03:00:00:, 06:000:00 and 09:00:00 respectively) and, from the second forecast, which starts at 12:00:00, we take the first, second, third and fourth time steps (12:00:00, 15:00:00, 18:00:00 and 21:00:00 respectively). The horizontal resolution was approximately 80 km prior to 2016-06-21 and approximately 40 km afterwards. Following its continuous development, the CAMS model has been upgraded every year with components that significantly change the UVI forecasts. For instance, improved handling of cloudiness (implemented on 2017-01-24) and a new extraterrestrial UV spectrum (implemented on 2017-09-26) have resulted in significant improvements (W. Wandji, 2018). As these changes only affect the data following the upgrade, the CAMS UV forecast is not a homogeneous time series but should be considered as an evolving UV product with which measurements can be compared. More information on the IFS and CAMS-IFS models is available at https://atmosphere.copernicus.eu/node/326. UVI-CAMS computed for clear-sky conditions will be compared against UVI-RADIO or UVI-BENTHAM measured in clear-sky conditions and UVI-CAMS computed for all-sky conditions will be compared against UVI-RADIO or UVI-BENTHAM measured in clear-sky conditions.

The TEMIS UVI product is derived from the TOC at solar noon obtained by satellites (SCIAMACHY, GOME-2A or GOME-2B depending on the time period. Following the parameterization of Allaart et al. (2004), a first estimate of UVI is computed with the TOC. The final UVI under clear sky at local solar noon (called UVI-TEMIS hereafter) is obtained by correcting the first estimates for the effects of the Earth-Sun distances, surface albedo, elevation and aerosols (Zempila et al., 2017). The spatial resolution is 0.5 x 0.5°.

### 4.1.2 Comparison results

Ground-based UVI measurements from the UV-Indien Network were compared with satellite estimates and model UVI estimates taking the nearest model grid point or satellite pixel for each station. Then, for each satellite or model estimate computed at this point, we looked for the closest UVI-BENTHAM or UVI-RADIO measurement. Finally, for each pair of measurements, we checked the following requirements: the time difference between the estimate and the reference had to be less than 5 minutes and the SZA difference had to be less than 5 degrees. For these comparisons, the UVI-BENTHAM was also scaled up and linearly interpolated at a resolution of 5 min. For data from the OMUVBG and TROPOMI product, UVI estimates located within 10 km of the corresponding ground-based station we selected. For CAMS, the closest grid point available for each

station was taken: 14.8, 9.6, 7.2 and 4.7 km from Antananarivo, Anse Quitor, Mahé and St-Denis respectively. For TEMIS, the closest grid point available for each station was used: 7.8, 13.0, 12.1 and 11.8 km from Antananarivo, Anse Quitor, Mahe and St-Denis respectively. Here, it was chosen to compare the satellite product or model forecast with results from the four stations of the UV Indien Network that have the longest time periods covered (Antananarivo, Anse Quitor, Mahé and St-Denis).

Figure 2 shows the period covered by each data set at these four stations. BENTHAM has been collecting data since 2009 but, in this study, only the period between 2016 and the present was chosen because the radiometers were installed only in 2016 or later (in 2017) depending on the stations. The Pearson correlation coefficient and absolute and relative differences were calculated for each ground-based instrument installed at each station. Then, the means and medians of the absolute and relative differences between ground-based measurements and the satellite or model estimates were computed (called Mean-

AD, Median-AD, Mean-RD and Median-RD hereafter). The correlation between the BENTHAM at St-Denis and the other measurements is shown in Figure 3. All data sets are well correlated with the UVI-BENTHAM under clear sky conditions; Pearson correlation coefficients are greater than 0.9, except for OMUVBG (0.55) at Saint-Denis. Correlation results for the radiometers at St-Denis, Antananarivo, Mahé and Anse Quitor stations are available in the appendix (Figure A1, A3, A5 and A7 respectively). Correlation coefficients are greater than 0.8 at all stations except Anse Quitor (for TEMIS and OMUVBG),

Mahé (for OMUVBG, GOME and TEMIS) and Saint-Denis (OMUVBG). On average, for all stations combined, OMUVBG is the product with the lowest correlation coefficients while CAMS shows the highest correlation coefficients. Mahé is the station that is least well represented by most models, with correlation coefficients between 0.19 and 0.9. The absolute and relative differences between satellite product or forecast model and UVI-BENTHAM at St-Denis are shown in Figure 4 and Table 4. The clear-sky conditions are presented in blue on the boxplot (Figure 4) and the corresponding values are reported

in Table 4. The all-sky conditions are presented in red on the boxplot (Figure 4) and the corresponding values are shown in brackets in Table 4. Looking at the median of the absolute differences (Median-AD) under all-sky conditions, it can be seen that all data sets, with the exception of OMUVBG, overestimate UVI relative to the UVI-BENTHAM measurements; median overestimation ranges between 0.21 (TROPOMI) and 0.96 (TEMIS). In clear-sky conditions, the median absolute differences decrease but a UVI overestimation can still be observed for CAMS (+0.21) and TEMIS (+0.27). The smallest median differ-

ence is obtained for GOME (-0.01). UVI-RADIO is expected to be aligned with UVI-BENTHAM since it has been recalibrated with the UVI-BENTHAM measurements, so it will not be discussed further here. The mean of the absolute differences under all-sky conditions ranges from $0.34 \pm 2.54$ (OMUVBG) to $+2.31 \pm 3.26$ (TEMIS). Under clear-sky conditions, this absolute difference is smaller and ranges between $-0.25 \pm 0.73$ (TROPOMI) and $0.47 \pm 0.55$ (CAMS). The Mean-RD under all-sky conditions is between $23.93 \pm 82$ % (OMUVBG) and $+59.73 \pm 164\%$ (TEMIS). These means and standard deviations are

strongly reduced under clear sky conditions; they are between $0.76 \pm 8.66$ % (GOME) and $12.33 \pm 12.27$ (CAMS). Similarly, the Median-RD under all-sky conditions are higher than those obtained under clear-sky conditions. UVI estimates by satellite or models are closer to instrument measurements made on the ground in clear-sky conditions. This is mainly due to the spatial resolution and representation of clouds. The satellite pixel or model grid point is representative of a 5.6 x 3.7 km region for the best defined satellite (TROPOMI) or a 0.5 deg x 0.5 deg region (55 km x 55 km) (GOME, OMI and TEMIS). Thus the cloud

cover considered is representative of this entire region but it is not necessarily that directly observed above the ground-based

instruments. In addition, the pixels and grid points selected are not perfectly centred on the ground instruments. Finally, the four study sites are located in the tropics, present non-uniform topographic conditions, and are very close to the sea (with the exception of Antananarivo). These conditions favour the rapid development of clouds and complicate the estimation of cloud cover over the site by the satellite (Lakkala et al., 2020). Thus, the clear-sky conditions observed on the ground may not be

the same as those observed by satellites or models. This will induce a discrepancy between UVI derived from satellite and modelling and UVI observed at the ground. For the other stations (Appendix A1 to A8), all satellites and models underestimate the surface UVI (UVI-RADIO) except CAMS at Antananarivo, where UVI-CAMS is just above UVI-RADIO with a mean absolute difference of +0.1. In clear-sky conditions, the Median-AD vary between -0.01 for CAMS in Antananarivo and -6.0 for GOME in Mahé. In all-sky conditions, the Median-AD vary between 0.05 for CAMS in Antananarivo and -4.5 for GOME

in Mahé. In clear-sky conditions, the Median-RD vary between -0.5 % for CAMS in Antananarivo and -34.5 % for TROPOMI in Mahé. In all-sky conditions, the Median-RD vary between 15.8 % for CAMS in St-Denis and -32.7 % for TROPOMI in Mahé. In clear-sky conditions, the Mean-AD vary between 0.1 for CAMS in Antananarivo and -5.8 for GOME in Mahé. In all-sky conditions, the Mean-AD vary between 0.2 for CAMS in Antananarivo and -3.5 for TROPOMI and GOME in Mahé. In clear-sky conditions, the Mean-RD vary between -10.9 % for CAMS in Antananarivo and 56.15% for TEMIS in Anse Quitor.

In all-sky conditions, the Mean-RD vary between -25.0% for TROPOMI in Mahé and 26.4 % for TEMIS in Mahé. Although TROPOMI has large relative differences, especially in Mahé, it is also the product with the most consistent differences. Its relative and absolute standard deviation is the smallest at all stations. It should be noted that all satellite or model estimates give poor results at the Mahé station. An examination of the percentage of clear days shows that, in Mahé, Anse Quitor and Antananarivo, there are only 16% clear days while, in St-Denis, there are 27% clear days. In addition, Anse Quitor and An-

tananarivo radiometers have a 1 min resolution, while those in Mahé and St-Denis have a 5 min resolution. The small number of clear-sky days combined with the larger temporal resolution, reduces the number of points compared in clear-sky conditions at Mahé. For OMUVBG, TROPOMI, GOME and TEMIS, which produce only one UVI estimate per day, the number of points compared ranges from 72 to 253. Mahé station is also closer to the equator than the other stations (at about 4° South) and is strongly influenced by the convection in this region. Outliers and large differences could be due to numerous issues. The

satellite measurements and the modelled UVI do not have spatial resolutions that can accurately represent the sky conditions just above the ground-based instruments. The satellite or model grid points used in this study are either the closest grid point to the station or the average of four grid points encircling the station location. Nonetheless, for satellite estimates or model results, spatial resolution can be as large as 50km (TEMIS or GOME) or as low as 3.7 km (TROPOMI). Also, cloud conditions, albedo or altitude can vary strongly over distances smaller than 100 km. These differences can affect the quality of the model forecast

or of the inversion algorithm used by the satellite. Differences between a satellite product or model forecast and radiometers measurements could also be explained by a drift in the radiometer calibration. Note that the radiometers installed in Mahé, Anse Quitor, and Antananarivo are still under the manufacturer's calibration and this could explain some of the observed differences with satellite or model estimates The recalibration of radiometers in Antananarivo, Anse Quitor and Mahé is planned for 2021. Nevertheless, UVI-BENTHAM, which provides high quality data and is recalibrated every 3 months, still shows a

Median-RD between -2.5% (TROPOMI) and 11.3% (CAMS), and a Median-AD between -0.2 (TROPOMI) and 0.3 (TEMIS),

in clear-sky conditions. Correlation, boxplot and tables for UVI-RADIO at St-Denis, Antananarivo, Mahé and Anse Quitor are available in the supplementary information (Appendix Figures A1 to A8 and appendix Tables A1 to A4).

## 4.2 Diurnal Variation of UVR and CF

For each station, the variability of the UV index, as measured by the radiometers, was analysed to understand the variability of the dataset and ensure its quality. The diurnal cycle of UVI was calculated from the start of measurements at each station until June 2020 for Anse Quitor, November 2020 for Mahé and December 2020 for Antananarivo and St-Denis. The results are presented in Figure 5: the mean UVI (blue line), the UVI maximum (continuous red line) and the first and third quartiles (blue shading) of the UVI in all-sky conditions during the day are represented for Antananarivo, Mahé, Anse Quitor and St-Denis. For a given station, in clear-sky conditions, the mean UVI is represented by a green line, the maximum UVI by a dashed red line and quartiles by shades of green. In all-sky conditions, the mean UVI at local solar noon varies from about 10 (Antananarivo, Anse Quitor, St-Denis) to about 14 (Mahé). The difference between Mahé and the other three stations can be explained by the latitudes of the stations: Antananarivo, Anse Quitor and St-Denis are at about 20° South while Mahé is at 4.6° South. Therefore, in Mahé, SZA are lower throughout the year, which induces a higher UVI during the year. For clear-sky conditions, the mean UVI is higher than for all-sky conditions at all stations. In addition, the first and third quartiles are closer to the mean in clear-sky conditions. This is due to the impact of cloud attenuation on UVI variability. It can be seen that the UVI maxima can be higher than 20 for Anse Quitor, Antananarivo and Mahé. For St-Denis, the UVI maxima can reach 16. The highest UVIs are observed at the Mahé station with maxima of up to 25. At all stations, the UVI maxima are higher for all-sky conditions than clear-sky conditions. These maxima of UVI in all-sky conditions, can be 1 to 4 units of UVI higher than maxima in clear-sky conditions. This is due to the enhancement of UVI by fractional cloud cover, which can produce multiple scattering, thus enhancing the UVI on the surface. This phenomenon was previously observed at St-Denis (Lamy et al., 2018) and has been described in past studies for other regions of the world (Badosa et al., 2014; Sabburg and Wong, 2000). The largest differences between the maxima in all-sky conditions and clear-sky conditions are observed at Mahé, the smallest differences are observed at St-Denis and Anse Quitor. Antananarivo also shows a significant absolute increase of UVI, by about 2 to 3 units. Antananarivo is at a higher altitude than the other three stations, at about 1.3 km asl. The city is also located inland and suffers from heavy pollution. Therefore, both aerosols and altitude can be expected to have a significant impact on the variability of surface UVIs. In Antananarivo, the peaks of UVI mean and maxima are aligned for both clear-sky and all-sky conditions. In St-Denis the peak of mean UVI occurs later in the day for clear-sky conditions than for all-sky conditions. In Anse Quitor, the peak of UVI is early in the day. This is due to the time zone used at Anse Quitor. Rodrigues Island, where Anse Quitor is located, shares the same time zone as St-Denis (Reunion island) or Mahé (Seychelles) but is farther east by about 8 degrees of longitude. In Mahé, the peaks of mean UVI, in all-sky and clear-sky conditions, are aligned. This is not the case for the maxima, which occur 1 to 2 hours earlier in all-sky conditions than in clear-sky conditions. The position of the UVI peaks can be explained by the variability of the cloud cover as will be shown in the next section. The monthly averages of the daily UVI maximum and the daily UVI at solar noon have also calculated. Although the datasets do not cover a sufficient period to be climatologically significant, the annual variability and the differences between UVI and UVI daily maxima at

solar noon present interesting results. Figure 6 shows these monthly averages for the four stations of Antananarivo, St-Denis, Mahé and Anse Quitor. The measurement periods used to obtain these monthly averages were the same as for Figure 5. The daily maxima of UVI are in red lines with filled circles, the corresponding sun zenith angles are in orange dashed lines with filled circles. The UVIs at local solar noon are marked with blue lines and crosses, and the corresponding sun zenith angles are shown as dashed orange lines with crosses. For all stations and all months of the year, the maximum daily UVI ($UVI_{DMAX}$) is

higher than the average UVI at solar noon ($UVI_{SNOON}$). These monthly averages are based on data sets for all types of cloud cover. Thus, the cloud cover will introduce a bias in the result. $UVI_{DMAX}$ include maxima of UVI during cloud enhancements, which could occur earlier or later than the local solar noon, as discussed in the previous result (Figure 5). UVIDMAX also includes maxima of UVI during clear-sky days, which will usually occur at local solar noon. $UVI_{SNOON}$ will include only UVI at local solar noon with or without cloud cover. This is why $UVI_{DMAX}$ is always higher than $UVI_{SNOON}$ in all-sky conditions.

The differences range between 1 during austral winter at St-Denis and 6 in austral summer at Mahé. The corresponding SZA differences range from 0 to 6 °. Looking at the few complete clear-sky days revealed a $UVI_{SNOON}$ and a $UVI_{DMAX}$ that were almost equivalent. However, as there were not enough clear sky measurements per month and per SZA bin, these results are not represented here.

## 4.3 Seasonal Distribution and Variation of UVR and CF

Early results from the cameras provide a brief description of the mean diurnal cloud cycles over each station. Figure 7a shows the mean diurnal cycle of the cloud fraction (CF) over Antananarivo for the entire year (black curve), along with the distribution of the first and third quartiles (blue shaded areas). Seasonal means of CF are also presented for December, January and February (DJF, in green), March, April and May (MAM, in red), September, October and November (SON, in orange) and June, July and August (JJA, in purple). The density of the corresponding data set for each month of the year is represented in Figure

7b. Figure 7c represents the difference of UVI maxima between clear-sky UVI and all-sky UVI for the same period. The UVI distribution is also available in Figure 6d. The same results are presented for St-Denis in Figure 8. Over Antananarivo (Figure 7), the annual mean diurnal cycle shows intermediate values of CF, of about 0.6 at the beginning of the day. CF tends to decrease during the day and to increase in the late afternoon. Antananarivo is located in a mountainous region in the highlands of Madagascar at 1370 m asl. The difference in CF in the four seasons is not statistically significant, especially for the JJA

and SON seasons. Since the camera at Antananarivo was installed in April, 2019, there are not yet significant numbers of data points for certain months of the year. Nonetheless, since the radiometer at Antananarivo was installed in 2016, there is a significant number of UVI data over the whole year (Figure 7d). The largest difference between maxima of UVI in clear-sky conditions and maxima of UVI in all-sky conditions can be observed for the DJF and JJA seasons (Figure 7c). CF diurnal profiles are quite different between the DJF and JJA seasons, with DJF showing CF higher than the annual mean while JJA

shows CF lower than the annual mean. The CF diurnal cycle alone is probably not the only factor involved in triggering UVI enhancement: cloud distributions and type of clouds probably play a role (Sabburg and Wong, 2000; Calbó et al., 2005; Badosa et al., 2014). The measurements proposed here will be fully useful in carrying out this type of study.

Over St-Denis the mean diurnal CF cycle shows a different profile from that of Antananarivo. CF values are lower in the morning, at around 0.4, and increase during the day to reach 0.6. St-Denis is located on the northern part of Reunion Island, a mountainous island under the influence of the south-east trade winds which induce cloud formation on the southern part of the island during the morning. During the day, the clouds tend to overflow over the rest of the island, thus inducing a rising CF during the day. The lowest values of CF throughout the day are observed in SON. DJF, MAM and JJA follow the annual mean. For this station, well spread CF and UVI data are available throughout the year (Figure 8b and 8d). A glance at the difference between the maxima of UVI in clear-sky conditions and the maxima of UVI in all-sky conditions (Figure 7c), shows that almost all the highest differences are observed for the DJF season. These differences can reach -2.4. The JJA season also shows large differences during the morning but, after 10 am, the differences are smalled than in DJF most of the time. CF also starts to increase faster from 10:00 a.m. onwards for the JJA season. (Figure 7a). This result could indicate that UVI enhancement occurs less frequently above a certain CF threshold. The cameras of the UV-Indien network present promising results for studying cloud variability and its impact on UV radiation in the Indian Ocean region. CF alone is probably not sufficient to give an understanding UVI enhancement. Cloud types, types of cloud distribution and solar occultation by clouds also need to be considered. Moreover, the UV Indien Network is just starting up, so there is not yet enough data for significant climatological studies to be conducted.

The statistics obtained on the diurnal cycles of the UVI show consistent results, the maxima measured by the radiometers do not seem to be outliers but rather the consequence of physical phenomena such as the increase of radiation by clouds. The monthly averages of daily UVI maxima and UVI measured at solar noon on the 4 sites of Antananarivo, St-Denis, Mahé and Anse Quitor are consistent with the variation of the zenith solar angle on these different sites. For the Antananarivo and St-Denis sites, coherent behaviour is observed between the all-sky camera measurements and the radiometer measurements over the whole data set or over the four seasons represented here.

## 5   Data Availability

Data from the UV radiometers and the total sky imagers can be downloaded from Zenodo and are referenced under the following doi: https://doi.org/10.5281/zenodo.4811488 (Lamy and Portafaix, 2021).

The data from the UV radiometers are also available on the WOUDC website at these addresses for the four stations studied here :

- Saint-Denis, La Réunion : https://woudc.org/data/stations/?id=530

- Mahé, Seychelles : https://woudc.org/data/stations/?id=207

- Anse Quitor, Mauritius : https://woudc.org/data/stations/?id=532

- Antananarivo, Madagascar : https://woudc.org/data/stations/?id=531

The TROPOMI surface UV radiation product used in this study is available at https://nsdc.fmi.fi/data/data_s5puv.php

The AC SAF (GOME/METOP) data can be downloaded through the website at https://acsaf.org

The TEMIS data can be downloaded through the website at http://www.temis.nl/uvradiation/UVindex.html

The OMUVBG data can be downloaded through the website at https://disc.gsfc.nasa.gov/datasets/OMUVBG_003

The CAMS data can be downloaded through the website at https://apps.ecmwf.int

## 6 Conclusions

The UV-Indien network is an emerging network dedicated to UV radiation and cloud cover in the western Indian Ocean region. This network offers numerous perspectives for studies, both on the variability of these parameters at each site, and for example, for the validation of climate models. The diversity of the instrumented sites provides a good representation of the different environmental conditions in the region. Our region is essentially made up of poor countries, whose access is difficult. The challenge of maintaining such a network in this region is mainly related to the regular calibration of the radiometers and also to the repair of the instruments when failures occur. A protocol including three types of calibration was established to guarantee the quality of the measurements while meeting logistical and financial constraints. A calibration plan has been established with frequencies of the order of 2 years, but logistical difficulties or health crises can modulate these forecasts. Another difficulty lies in the regular transfer of data to a centralising organisation. Nonetheless, the calibration protocol and post-processing of the data ensures the quality of the data before distribution to the Zenodo and WOUDC databases. In order to assess the validity of the measurements so that future users can use them with confidence, we compared these measurements with satellite estimates and model estimates of the UVI. In clear-sky conditions, the correlation coefficient between the satellite or model estimates and the ground-based measurements was greater than 0.9 at all stations except Mahé and for all datasets except OMUVBG. In all-sky conditions, the largest UVI median absolute difference between the satellite or model estimates and the ground-based instruments was -0.4 (TROPOMI), -1.4(TROPOMI), -2.3(GOME) and -6.0(GOME) at St-Denis, Antananarivo, Anse Quitor and Mahé respectively. In clear-sky conditions, the largest UVI median absolute difference between the satellites or model estimates and the ground-based instruments was 1.1(TEMIS), -1.3(TROPOMI), -1.7(GOME) and -4.6 (GOME) at St-Denis, Antananarivo, Anse Quitor and Mahé respectively. In all-sky conditions, the smallest UVI median difference between the satellites or model estimates and the measurements of ground-based instruments was 0.1 (CAMS and TEMIS), -0.1 (CAMS), -0.1 (CAMS) and 0.1 (CAMS) at St-Denis, Antananarivo, Anse Quitor and Mahé respectively. In clear-sky conditions, the smallest UVI median difference between the satellites or model estimates and the measurements of ground-based instruments was 0.02 (TROPOMI), 0.04 (OMUVBG), -0.1 (CAMS) and -0.4 (CAMS) at St-Denis, Antananarivo, Anse Quitor and Mahé respectively. Compared to ground measurements, UVI-CAMS showed the most consistent results at all stations in both all-sky and clear-sky conditions. At St-Denis, satellite and model estimates were usually found to overestimate UVI compared to ground-based instruments, while at Antananarivo, Anse Qutor and Mahé, satellite and model were found to underestimate ground-based measurements of UVI.

The largest discrepancies were observed at the Mahé station. Many reasons can be evoked to explain these differences. The satellite or model resolution may not be able to accurately determine the sky conditions above the station. As Mahé is under the influence of the sea-breeze, the formation of clouds is frequent late in the morning. From the ground, these clouds could be

seen to induce irradiance enhancements due to multiple scattering at their edges. From the satellite's point of view, only strong backscattering would be observed and an attenuation of the clouds would therefore be applied to the UV product. This is also consistent with the time of the satellite estimates (solar-noon or overpass), which is late in the morning when cloud formation is frequent which would induce backscattering. UVI enhancement also occurs frequently during this time (Figure 5). These two phenomena could explain both the observed high UVI and the discrepancy with satellite and model estimates. A drift in

the radiometer calibration could also explain part of the difference. In addition, Mahé presents a small number of clear-sky comparison points since the KZ radiometer temporal resolution is 5 min and there are only 16% of clear-sky days. St-Denis and Anse Quitor are mountainous islands also under the influence of trade winds, where there is frequent cloud formation in the late morning. A fractional cloud sky cover could induce UVI enhancements measured by the ground-based instruments while satellites and model could apply attenuation of the measured UVI instead. These two stations are located at about 20 degrees

south, while Mahé is located at about 5 degrees south. Therefore, the maxima of UVI are higher at Mahé and the resulting enhancement of UVI should also be stronger. As the phenomenon of cloud enhancement of the UV index can be observed in the radiometer data of the four stations presented in detail here, these data are suitable for the study of this phenomenon. To enable users to investigate the impact of clouds on UVI, all-sky cameras have recently been deployed at each station of the UV Indien network. In this article, we have also proposed a first estimate of the variability of UVI at these different sites, to give

users a general idea of this variability. In a region where UVI values are very high all year round, these data are also useful for monitoring the intensity of ultraviolet radiation in the Indian Ocean for health preventive studies or health impact studies. The monitoring of long-term UVI trends in the region will allow us to evaluate their evolution in the context of climate change and the possible acceleration of the Brewer-Dobson circulation leading to a decrease in the total ozone column in the tropics and thus an increase in ultraviolet radiation levels.

*Acknowledgements.* We thank the AC SAF project of EUMETSAT for providing the GOME-2/Metop offline surface UV products used in this paper. We also thank the Finnish Meteorological Institute for providing the TROPOMI UV products, and especially Jukka Kujanpää for product development. The authors thank Reuniwatt for the support provided during the installation of the total sky cameras at the different sites of the UV-Indien network. Finally, the authors would like to thank the PMOD/WRC for carrying out the calibration of the Radiometer UVS-E-T-150154 in St Denis/Moufia.

The UV-Indien programme is supported by the European Union through PO INTERREG V, by the Reunion Island Council and by the French Government. The BENTHAM spectroradiometer and UV radiometer at St Denis are managed by OPAR (Observatoire de Physique de l'Atmosphère de la Réunion) and OPAR activities are funded by the Université de La Réunion and CNRS. Measurements of Bentham spectroradiometers are also supported by CNES (within the French programme TOSCA).

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

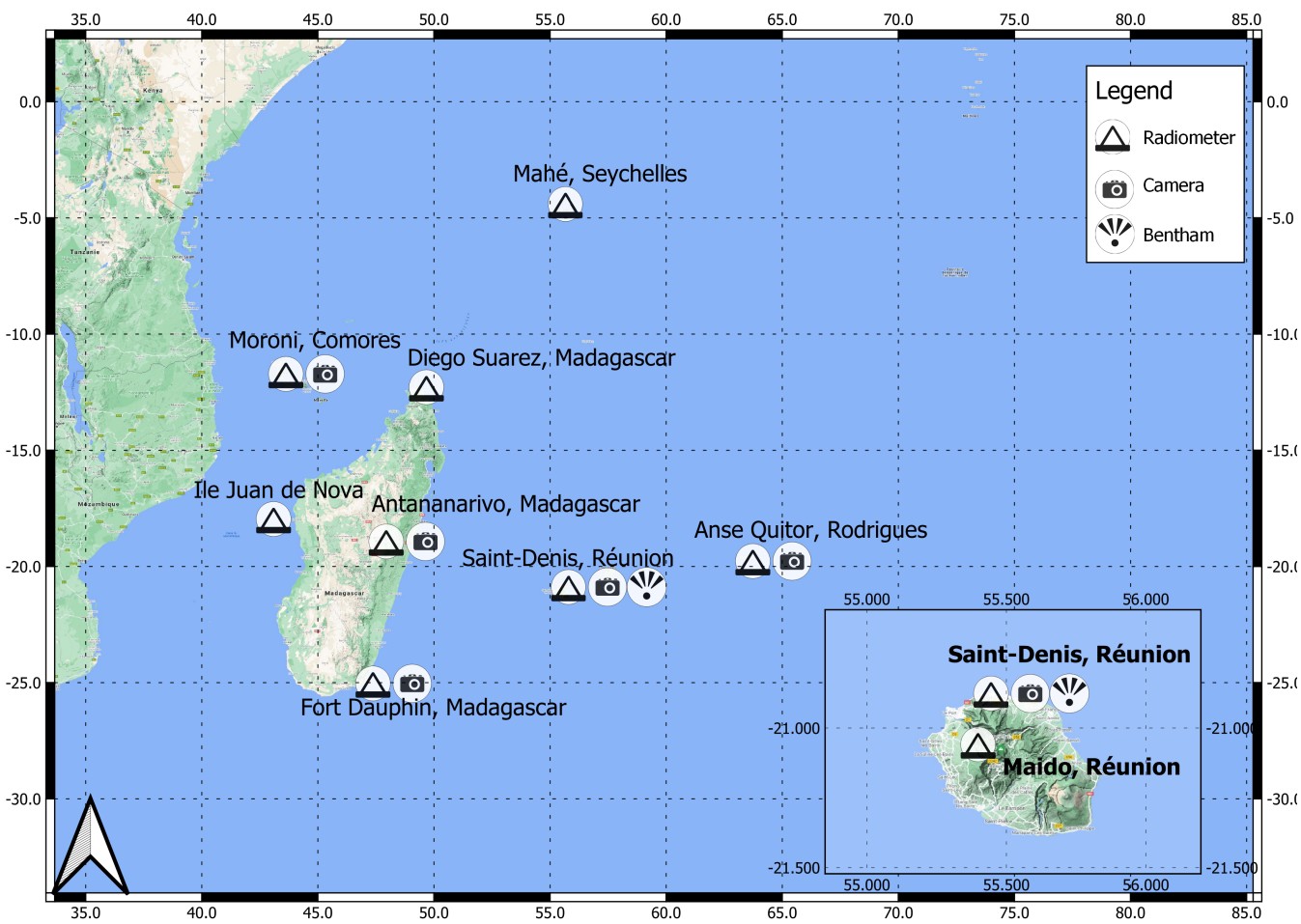

**Figure 1.** Location of the UV-Indien Stations

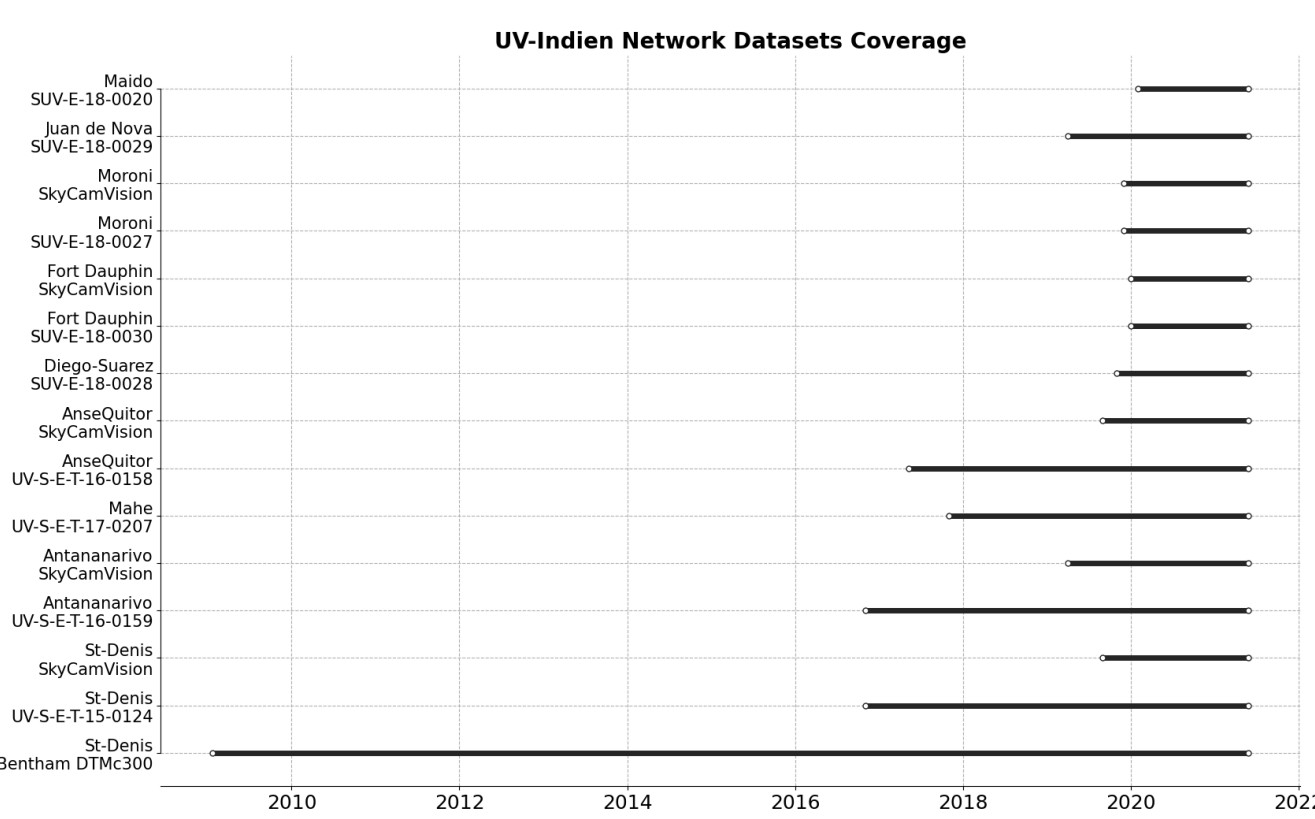

**Figure 2.** Timeline of the instruments of the UV-Indien network

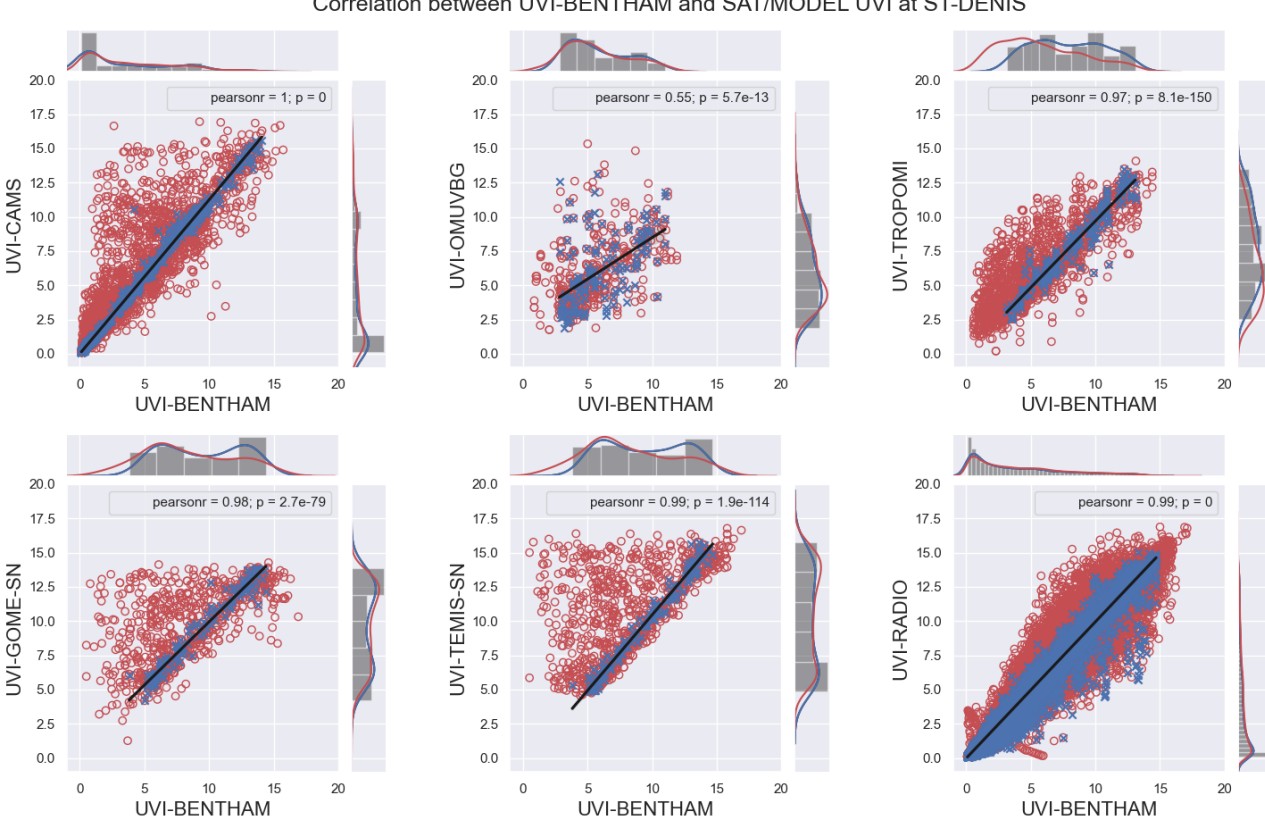

**Figure 3.** Correlation between UVI-BENTHAM and Sat/Model Estimates at ST-DENIS.

A histogram representing each dataset distribution is represented of the right and at the top side of each subfigure. The clear-sky measurements made by BENTHAM have been distinguished and are shown as blue crosses while the measurements for all-sky conditions are shown as red circles.

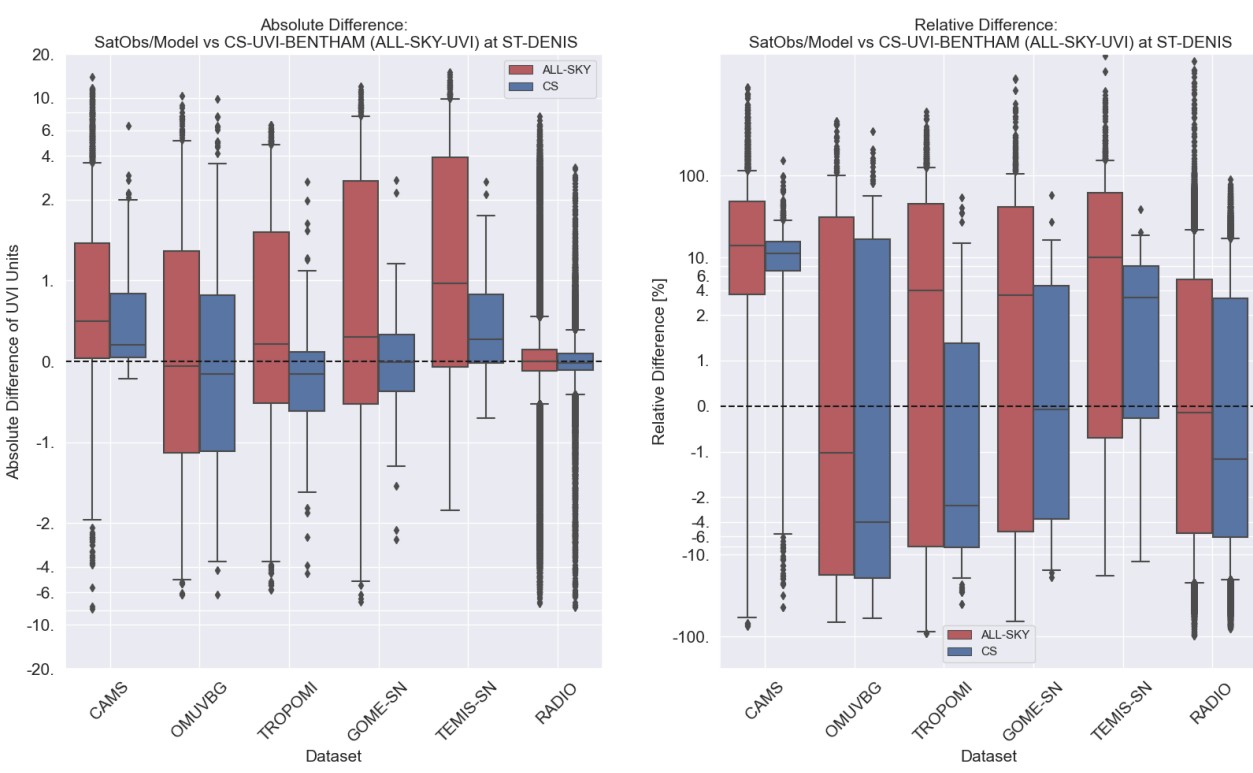

**Figure 4.** Boxplot of differences between UVI-BENTHAM and Sat/Model Estimates at ST-DENIS.

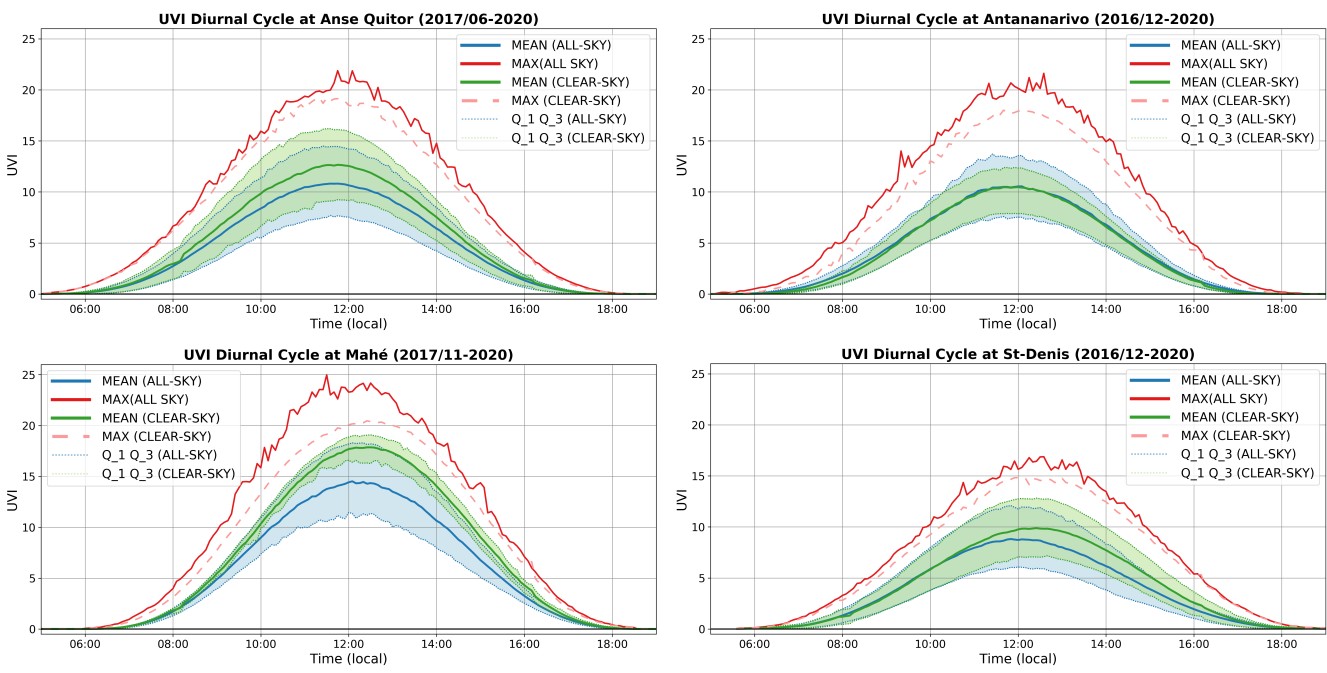

**Figure 5.** Diurnal Cycle of UVI at Anse Quitor, Antananarivo, Mahé and St-Denis.

In all-sky conditions: Mean UVI in blue, Max UVI in red, First and Third Quartile delimit the blue shaded area. In clear-sky conditions: Mean UVI in green, Max UVI in dashed-red, First and Third Quartile delimit the green shaded area.

# UVI Monthly Averages (All-Sky)

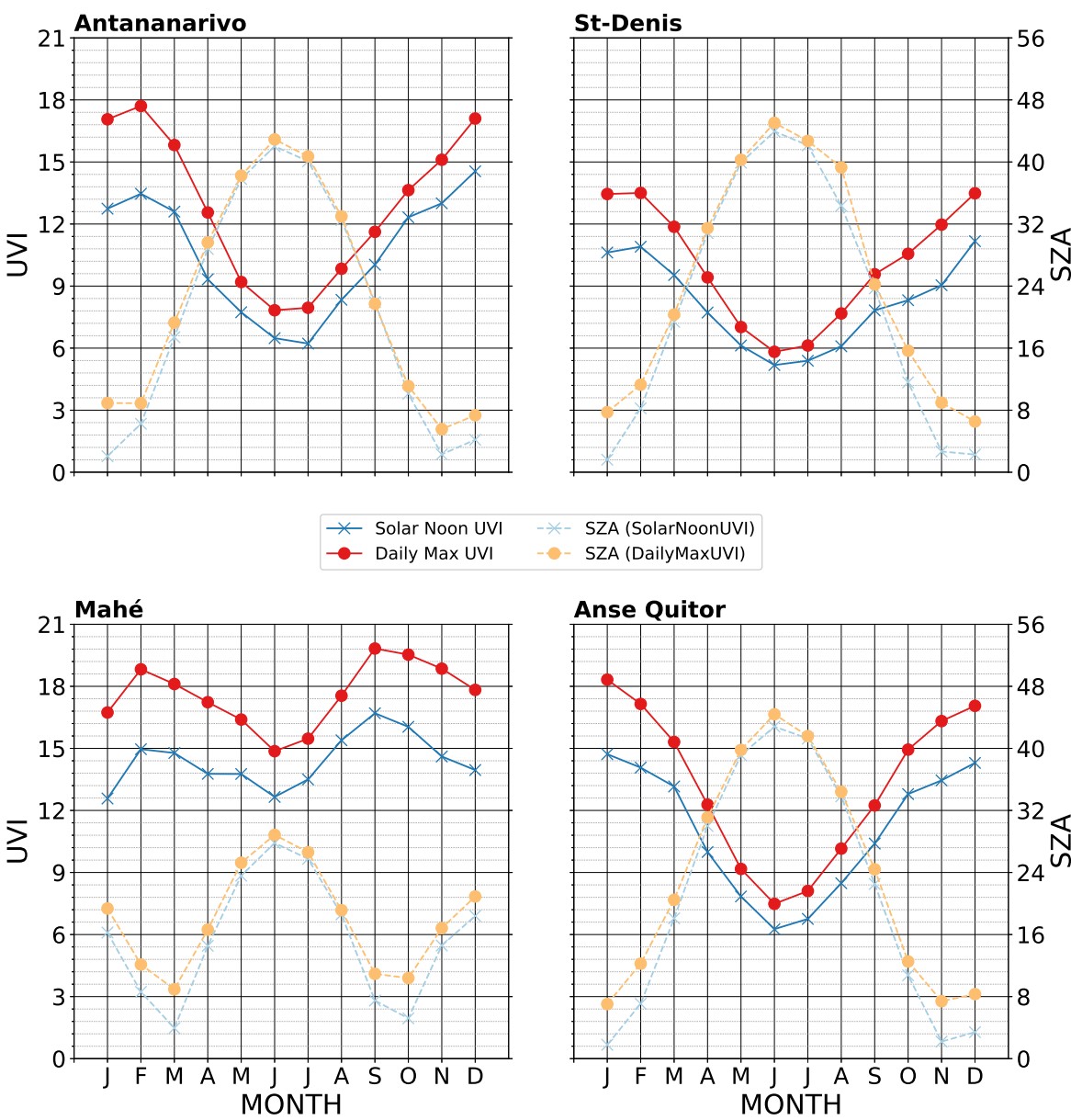

**Figure 6.** UVI Monthly Averages in all-sky conditions for the four stations of Antananarivo, St-Denis, Mahé and Anse Quitor. The monthly averages of the daily max UVI is represented by the red line with filled circles marker. The corresponding SZA is represented by the dotted orange line.

The monthly averages of the UVI at solar noon is represented by the blue line with blue cross. The corresponding SZA is represented by the blue dashed line.

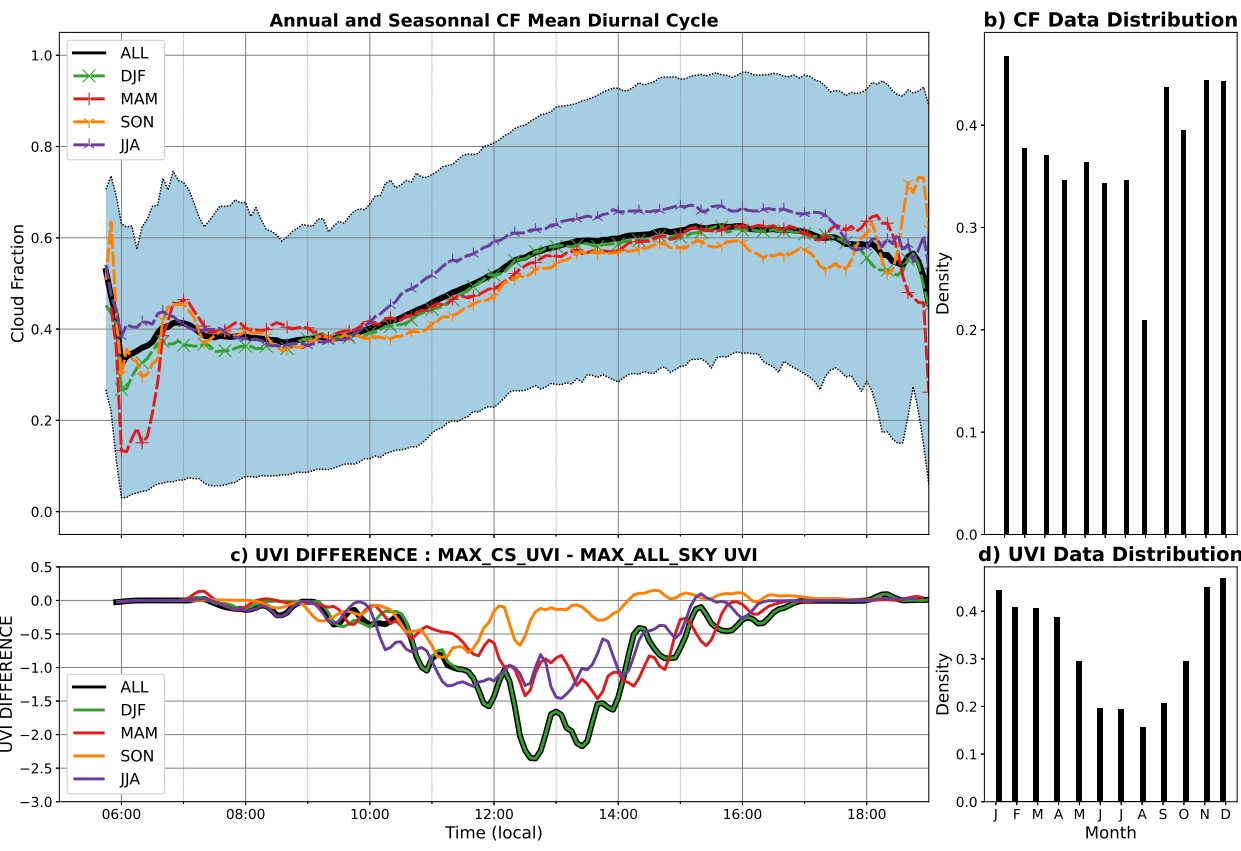

**Figure 7.** a) and b) Diurnal Cycle of CF at ANTANANARIVO. Annual Mean CF in black, first and third Quartiles delimit the blue shaded area. Seasonal Mean in green (DJF), red (MAM), orange (SON) and purple (JJA).

c) and d) Diurnal Cycle of the difference between maxima of UVI in clear-sky conditions and in all-sky conditions at ANTANANARIVO. Annual Max UVI differences in black. Seasonal Max UVI differences in green (DJF), red (MAM), orange (SON) and purple (JJA)

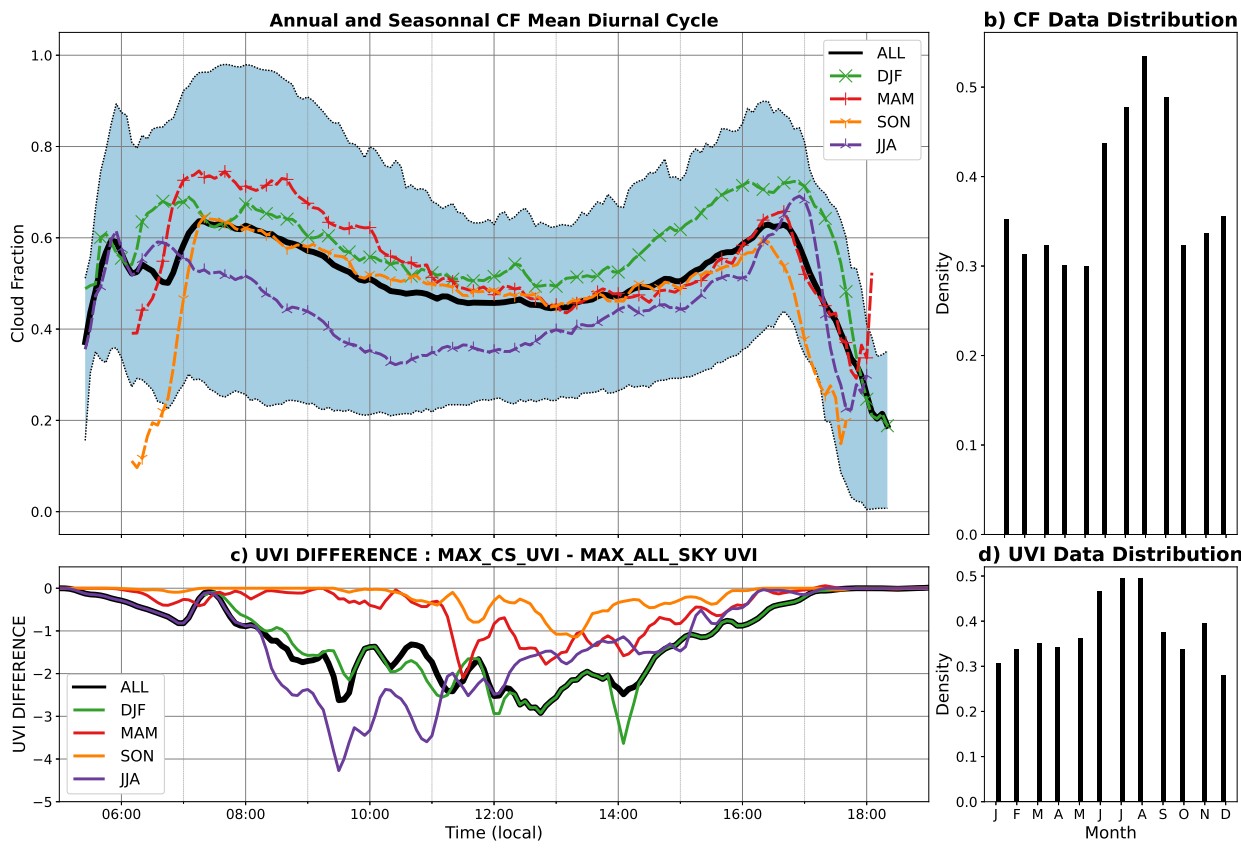

**Figure 8.** a) and b) Diurnal Cycle of CF at ST-DENIS. Annual Mean CF in black, first and third quartiles delimit the blue shaded area. Seasonal Mean in green (DJF), red (MAM), orange (SON) and purple (JJA).

c) and d) Diurnal Cycle of the difference between maxima of UVI in clear-sky conditions and in all-sky conditions at ST-DENIS. Annual Max UVI differences in black. Seasonal Max UVI differences in green (DJF), red (MAM), orange (SON) and purple (JJA)

| Station | Location | Coordinates | Instrument | Data since | Frequency | UTC+ | Altitude (asl) | Environmental conditions |
|---|---|---|---|---|---|---|---|---|
| Saint-Denis | Reunion | 20.902°S 55.485°E | BENTHAM DTMc 300 | 2009/01 | 15 min | +4 | 82m | Urban Coastal |
| | | | Radiometer KippZonen UVS-E-T 15-0124 | 2016/12 | 5 min | | | Low-Mid Pollution Tropical Climate |
| | | | All Sky Camera SkyCamVision | 2019/09 | 1 min | | | |
| Maido | Reunion | 21.079°S 55.383°E | Radiometer KippZonen SUV-E 18-0020 | 2020/02 | 1 min | +4 | 2200m | Rural Mountain |
| Anse Quitor | Rodrigues | 19.758°S 63.368°E | Radiometer KippZonen UVS-E-T 16-0158 | 2017/06 | 1 min | +4 | 32m | Rural Coastal |
| | | | All Sky Camera SkyCamVision | 2019/09 | 1min | | | No Pollution Dry Tropical Climate |
| Mahé | Seychelles | 4,679 °S 55.531 °E | Radiometer KippZonen UVS-E-T 17-0207 | 2017/11 | 5 min | +4 | 15m | Semi-Urban Coastal No Pollution Tropical Climate |
| Antananarivo | Madagascar | 18.916°S 47.565°E | Radiometer KippZonen UVS-E-T 16-0159 | 2016/12 | 5 min | +3 | 1370m | Urban Plateaux |
| | | | All Sky Camera SkyCamVision | 2019/04 | 1 min | | | Mid/High Pollution Tropical Climate |
| Diego Suarez | Madagascar | 12.279°S 48.287°E | Radiometer KippZonen SUV-E 18-0028 | 2019/11 | 1 min | +3 | 35m | Semi-Urban Coastal No Pollution Tropical Climate |
| Fort Dauphin | Madagascar | 25.028°S 46.995°E | Radiometer KippZonen SUV-E 18-0030 | 2020/01 | 1 min | +3 | 10m | Semi-Urban Coastal |
| | | | All Sky Camera SkyCamVision | 2020/01 | | | | No Pollution Tropical Climate |
| Moroni | Comores | 11.708°S 43.247°E | Radiometer KippZonen SUV-E 18-0027 | 2019/12 | 1 min | +3 | 12m | Semi-Urban Coastal |
| | | | All Sky Camera SkyCamVision | | | | | Low Pollution Tropical Climate |
| Ile Juan de Nova | France | 17.054°S 42.711°E | Radiometer KippZonen SUV-E 18-0029 | 2019/04 | 1 min | +3 | 10m | Small Desert Island Coastal No Pollution Tropical Climate |

**Table 1.** Stations of the UV-Indien Project

| Dataset | Platform | Type | Resolution | Field used for computation/calibration | | References |
|---|---|---|---|---|---|---|
| | | | | Ozone field used | Aerosol field used | |
| BENTHAM | Ground-Based | Spectrometer | dt = 15min | None | None | Brogniez et al. (2016) |
| Radiometer K&Z (UVI-RADIO) | Ground-Based | Radiometer | dt = 1 or 5 min | SAOZ or OMI | None | Cadet et al. (2020) |
| OMUVBG | Satellite | Spectrometer (Measurement at 360nm then Table Look-Up) | Daily (Overpass/Solar Noon) 0.25 x 0.25 deg. | OMI | OMI Krotkov et al. (1998) Herman et al. (2009) | Tanskanen et al. (2006) Arola et al. (2009) Levelt et al. (2006) |
| TROPOMI | Satellite | Spectrometer (Measurement at 354nm then Table Look-Up) | Daily (Overpass/Solar Noon) 5.6 x 3.7 km | TROPOMI L2 total ozone column product Garane et al. (2019). | Aerosol Climatology Kinne et al. (2013) | Lindfors et al. (2018) Lakkala et al. (2020) |
| GOME | Satellite | Spectrometer | Daily (local solar noon) 0.5 x 0.5 deg. | GOME-2A TOZ | MODIS Level 3 | Kujanpää and Kalakoski (2015) |
| CAMS | Model | Modelling | dt = 6 hours 40km (after 2016-06-21) 80 km (before 2016-06-21) | Modelled | Modelled | Bozzo A. (2015) |
| TEMIS | Model | Modelling | Daily (local solar noon) 0.5 x 0.5 deg. | SCIAMACHY GOME-2A MSR | No direct correction | Zempila et al. (2017) |

**Table 2.** Satellites, Model and ground-based data.

| Radiometer | Current Calibration Type | Date of Current Calibration | Next Calibration Date | Current Location |
|---|---|---|---|---|
| UVS-E-T 15-0124 | Bentham | 07-10/2019 | 05/2021 | Saint-Denis |
| UVS-E-T 16-0159 | Manufacturer | 12/2016 | 06/2021 | Antananarivo |
| UVS-E-T 16-0158 | Manufacturer | 06/2017 | 06/2021 | Anse Quitor |
| UVS-E-T 17-0207 | Manufacturer | 11/2017 | 06/2021 | Mahé |
| SUV-E 18-0020 | Bentham | 11/2019 | 03/2022 | Maido |
| SUV-E 18-0028 | Bentham | 07/2019 | 03/2022 | Diego Suarez |
| SUV-E 18-0030 | Bentham | 06/2019 | 03/2022 | Fort Dauphin |
| SUV-E 18-0027 | Bentham | 08/2019 | 03/2022 | Moroni |
| SUV-E 18-0029 | Bentham | 06/2019 | 03/2022 | Juan de Nova |

**Table 3.** Radiometers of the UV-Indien Network.

| STATS | SAT/MODEL Comparison Against CS-UVI-BENTHAM (ALL-SKY-UVI BENTHAM) at ST-DENIS | | | | | |
|---|---|---|---|---|---|---|
| | **UVI-CAMS** | **UVI-OMUVBG** | **UVI-TROPOMI** | **UVI-GOME-SN** | **UVI-TEMIS-SN** | **UVI-RADIO** |
| **MEAN-AD** | $0.47 \pm 0.55$ | $0.09 \pm 2.36$ | $-0.25 \pm 0.73$ | $-0.01 \pm 0.68$ | $0.40 \pm 0.62$ | $-0.03 \pm 0.39$ |
| | $(1.09 \pm 1.97)$ | $(0.34 \pm 2.54)$ | $(0.60 \pm 1.83)$ | $(1.23 \pm 2.86)$ | $(2.31 \pm 3.26)$ | $(0.02 \pm 0.65)$ |
| **MEAN-RD** | $12.33 \pm 12.27$ | $8.54 \pm 53.93$ | $-3.13 \pm 10.09$ | $0.78 \pm 8.66$ | $3.61 \pm 7.12$ | $-2.95 \pm 12.16$ |
| | $(48.36 \pm 105.87)$ | $(23.95 \pm 82.00)$ | $(30.85 \pm 70.85)$ | $(40.31 \pm 123.55)$ | $(59.73 \pm 164.38)$ | $(0.32 \pm 23.77)$ |
| **MEDIAN-AD** | 0.21 | -0.15 | -0.16 | -0.01 | 0.27 | -0.02 |
| | ( 0.50) | ( -0.06) | ( 0.21) | ( 0.30) | ( 0.96) | ( -0.00) |
| **MEDIAN-RD** | 11.32 | -3.98 | -2.52 | -0.07 | 3.27 | -1.17 |
| | ( 14.20) | ( -1.03) | ( 3.98) | ( 3.48) | ( 10.15) | ( -0.14) |
| **NDATA** | 1038 | 145 | 254 | 120 | 142 | 34665 |
| | (3069) | (408) | (1555) | (750) | (919) | (93300) |

**Table 4.** Mean and Median Absolute and Relative Differences between CS-UVI Sat/Model Measurements and CS-UVI-BENTHAM at ST-DENIS. Mean and Median Absolute and Relative Differences between ALL-SKY-UVI Sat/Model Measurements and ALL-SKY-UVI-BENTHAM at ST-DENIS in parentheses

 **Appendix A**

**A1**

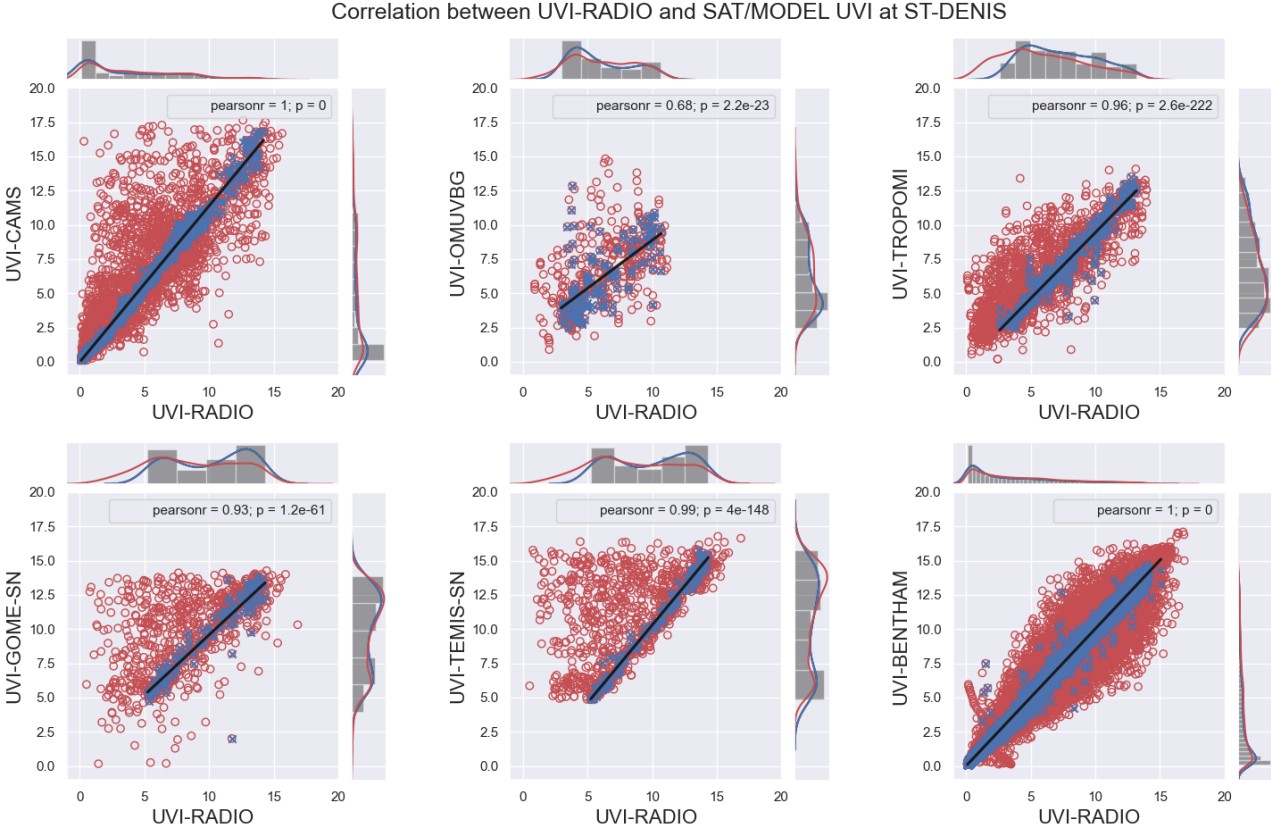

**Figure A1.** Correlation between UVI-RADIO and Sat/Model Estimates at ST-DENIS

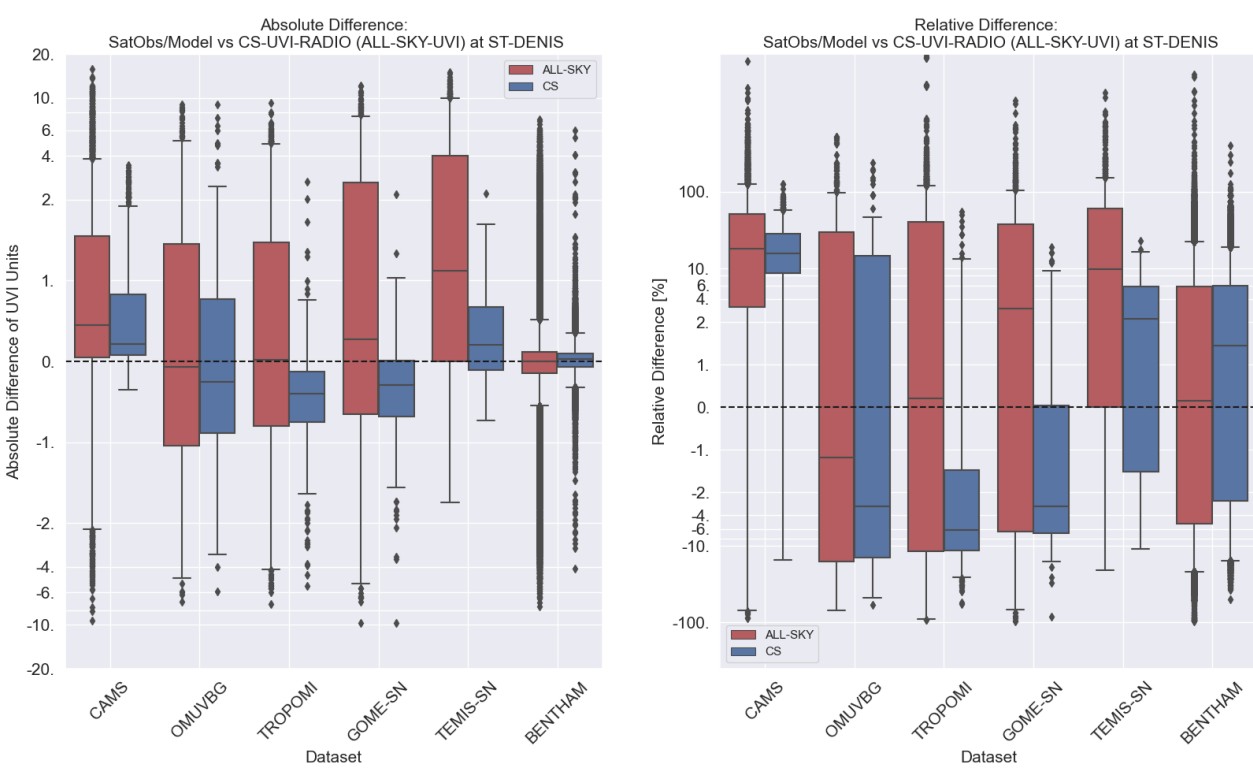

**Figure A2.** Boxplot of differences between UVI-RADIO and Sat/Model Estimates at ST-DENIS.

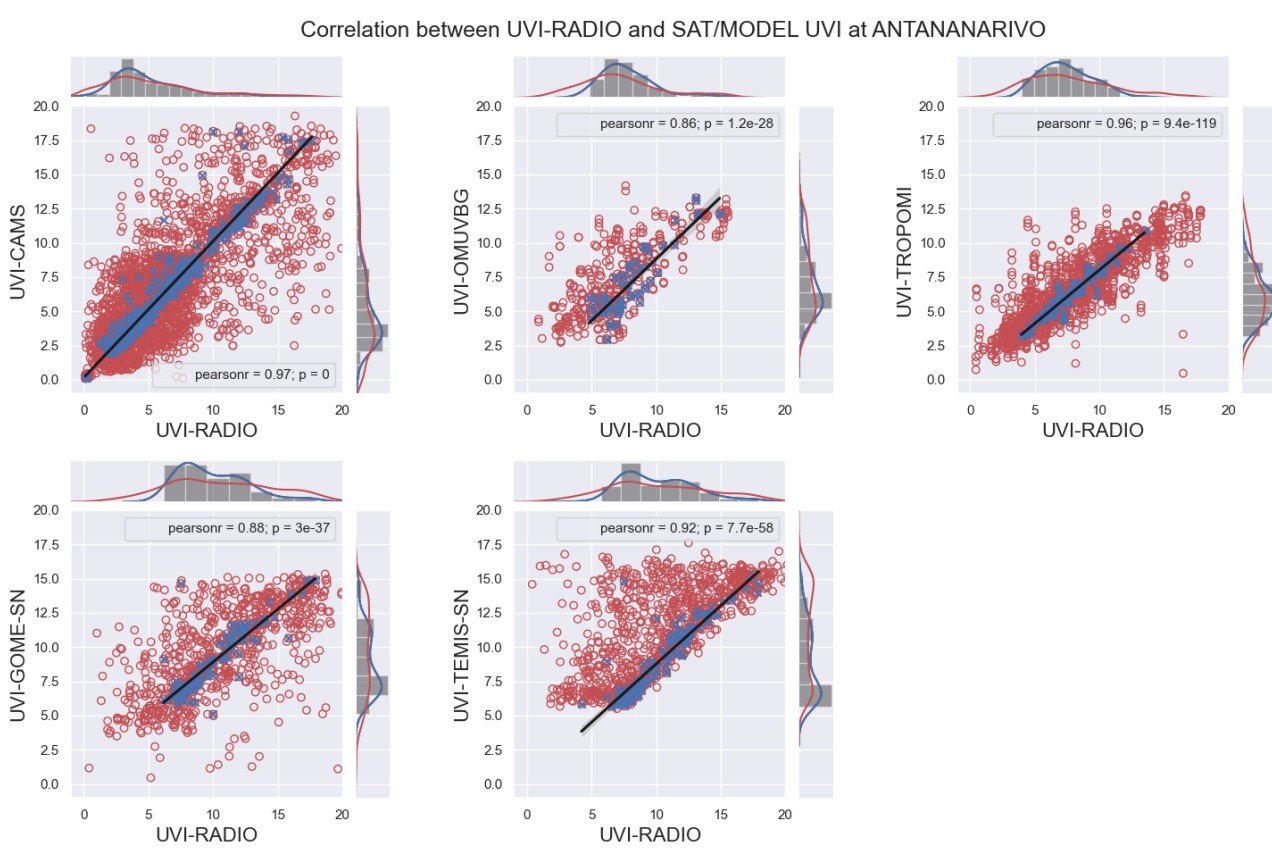

**Figure A3.** Correlation between UVI-RADIO and Sat/Model Estimates at ANTANANARIVO.

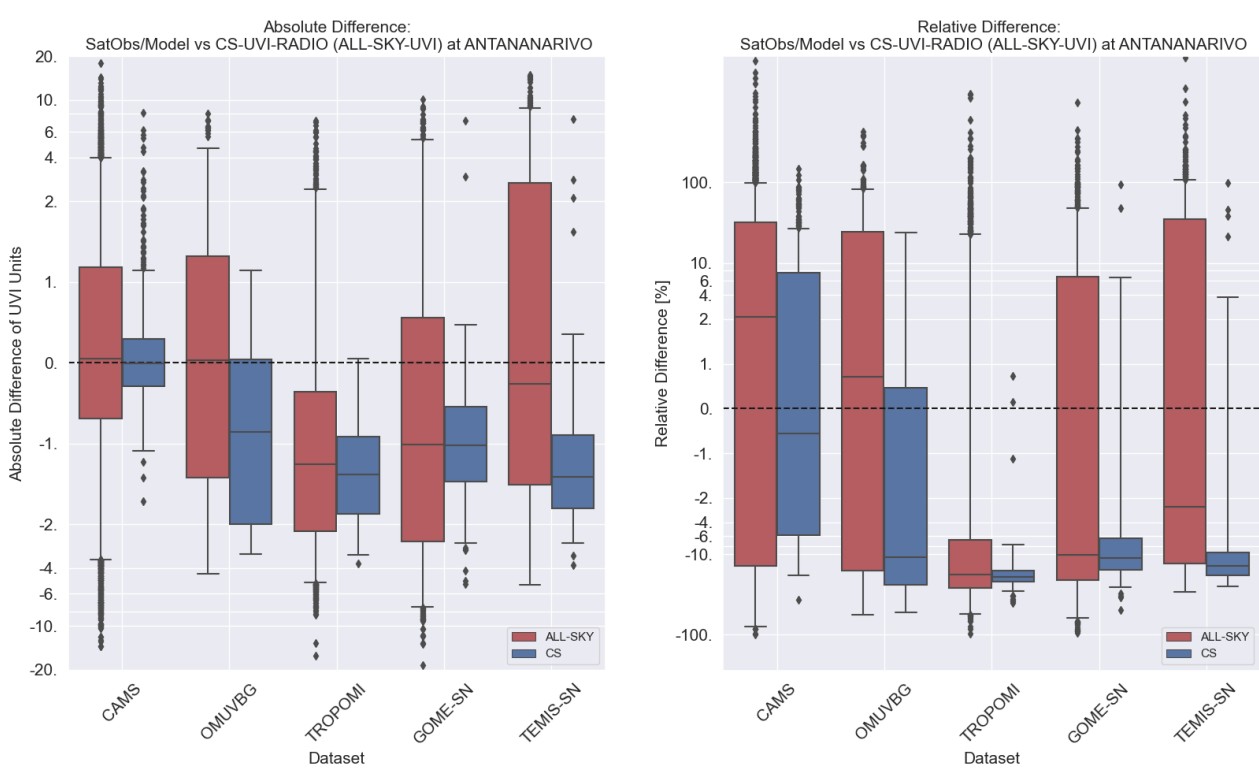

**Figure A4.** Boxplot of differences between UVI-RADIO and Sat/Model Estimates at ANTANANARIVO.

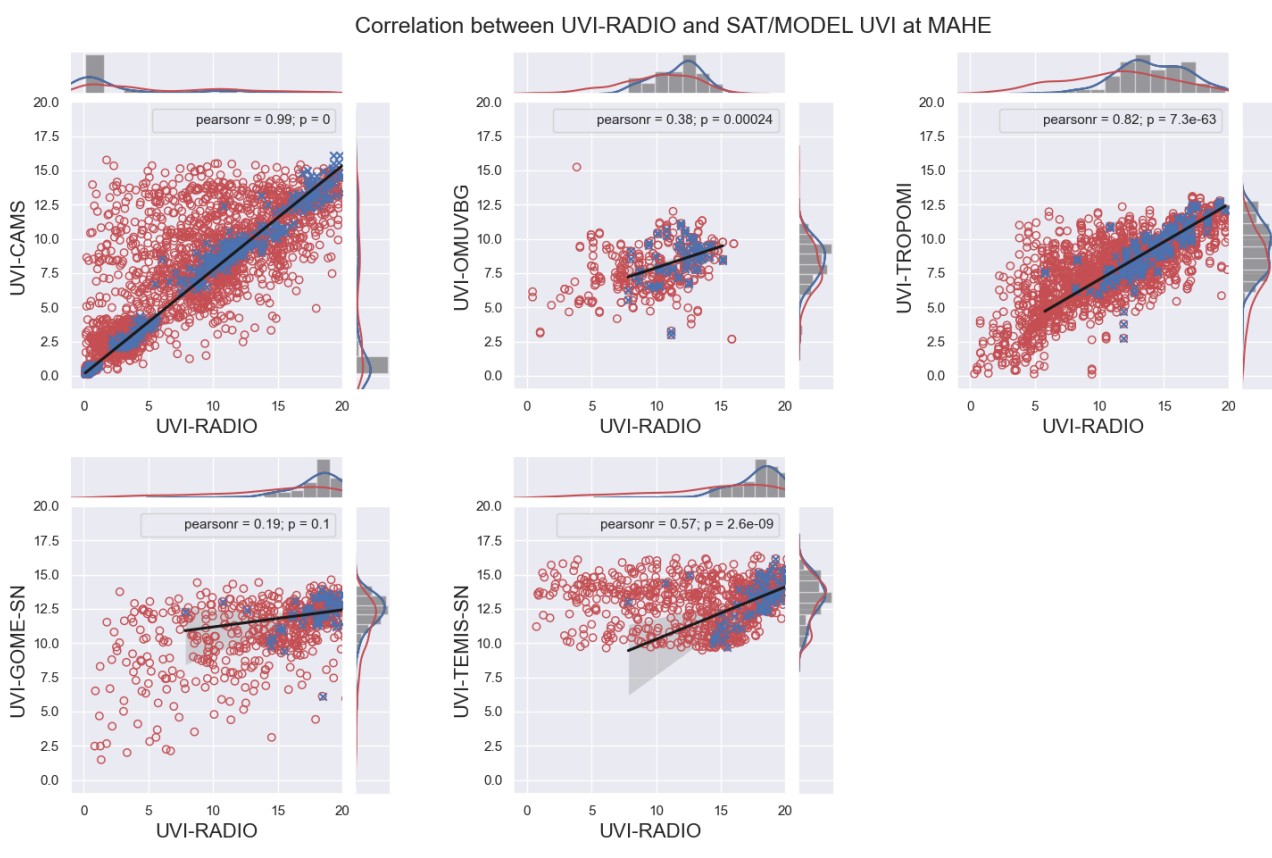

**Figure A5.** Correlation between UVI-RADIO and Sat/Model Estimates at MAHE.

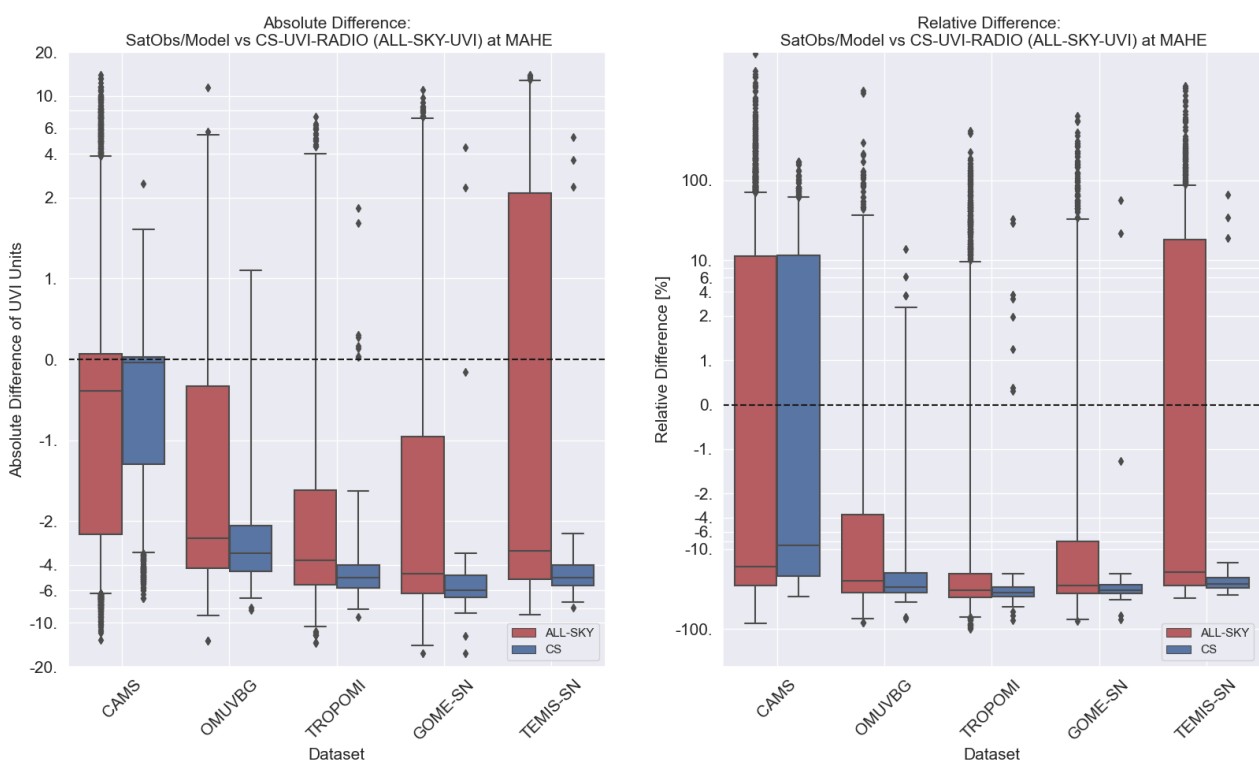

**Figure A6.** Boxplot of differences between UVI-RADIO and Sat/Model Estimates at MAHE.

Correlation between UVI-RADIO and SAT/MODEL UVI at ANSE QUITOR

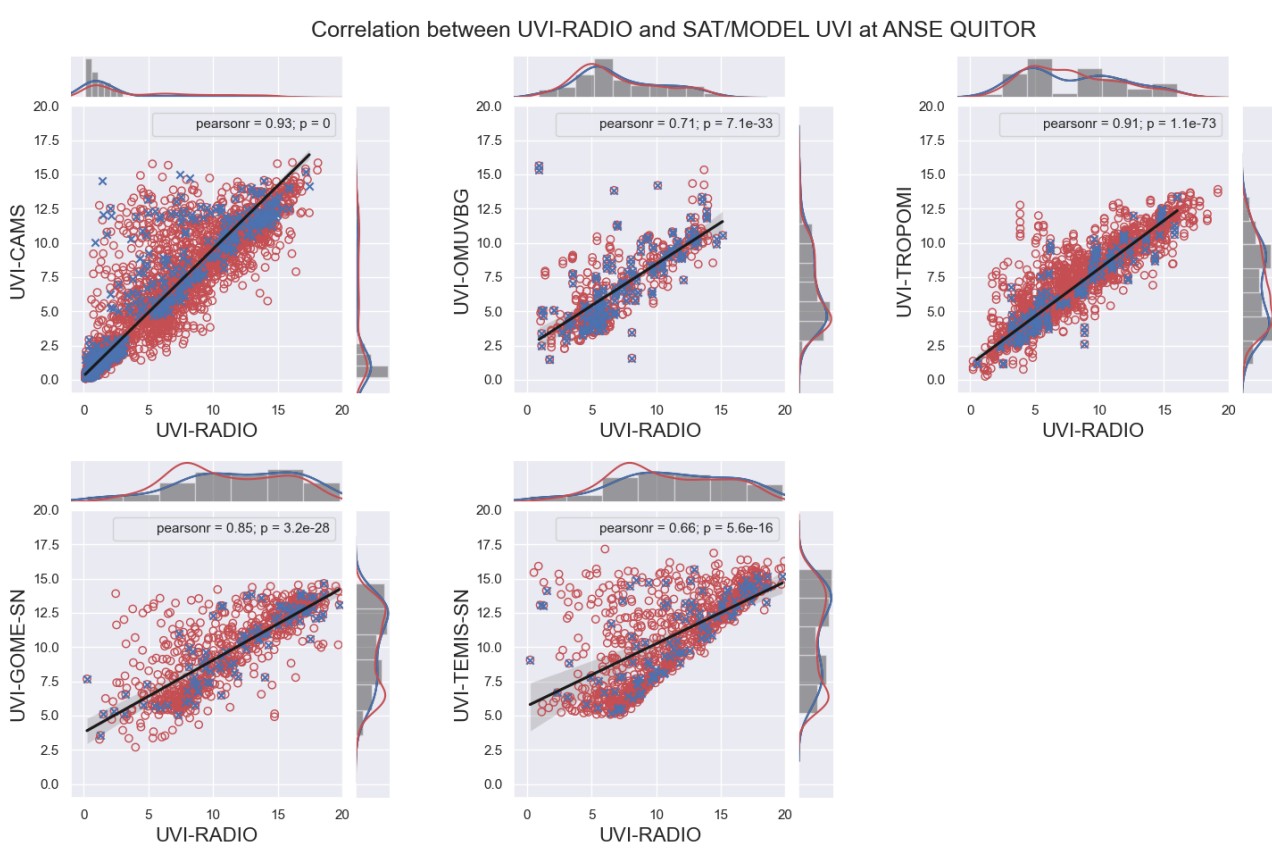

**Figure A7.** Correlation between UVI-RADIO and Sat/Model Estimates at ANSE QUITOR.

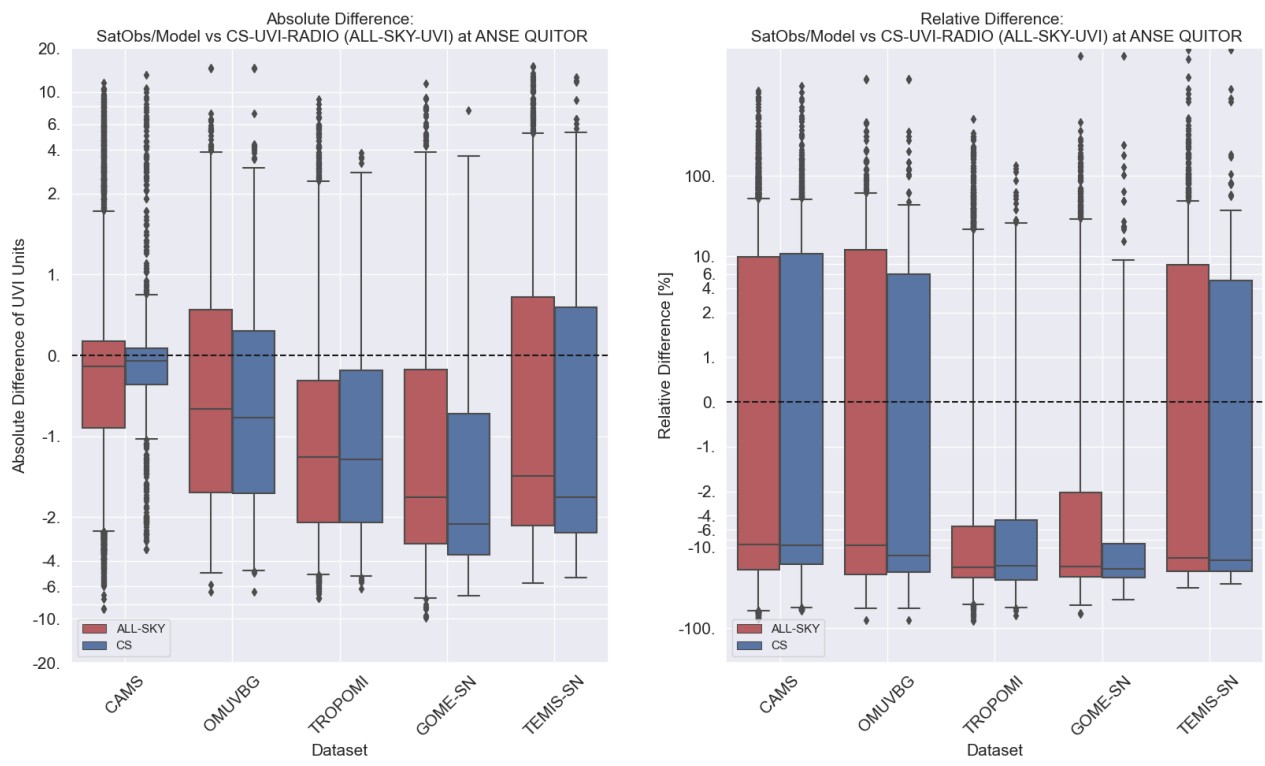

**Figure A8.** Boxplot of differences between UVI-RADIO and Sat/Model Estimates at ANSE QUITOR.

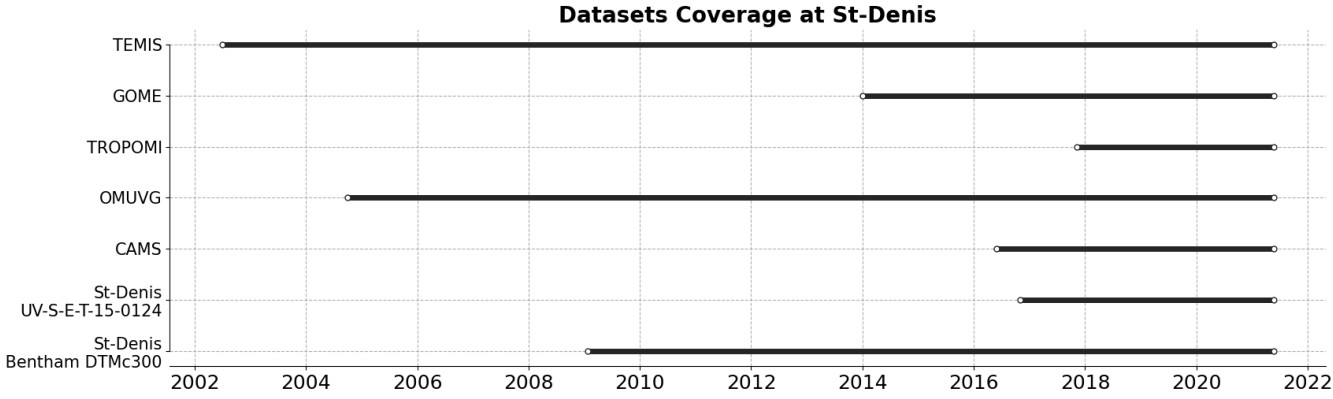

**Figure A9.** Timeline of the instruments of the UVI dataset used in the comparison at St-Denis.

| STATS | Comparison of SAT/MODEL with CS-UVI-RADIO (ALL-SKY-UVI RADIO) at ST-DENIS | | | | | |
|---|---|---|---|---|---|---|
| | UVI-CAMS | UVI-OMUVBG | UVI-TROPOMI | UVI-GOME-SN | UVI-TEMIS-SN | UVI-BENTHAM |
| **MEAN-AD** | $0.51 \pm 0.63$ | $0.07 \pm 1.89$ | $-0.46 \pm 0.75$ | $-0.43 \pm 1.10$ | $0.29 \pm 0.58$ | $0.02 \pm 0.22$ |
| | $(1.08 \pm 2.19)$ | $(0.38 \pm 2.47)$ | $(0.39 \pm 1.95)$ | $(1.09 \pm 2.92)$ | $(2.39 \pm 3.24)$ | $(-0.02 \pm 0.64)$ |
| **MEAN-RD** | $20.12 \pm 18.11$ | $6.22 \pm 41.58$ | $-6.45 \pm 10.87$ | $-3.53 \pm 9.84$ | $1.80 \pm 6.12$ | $4.17 \pm 13.00$ |
| | $(51.13 \pm 140.84)$ | $(21.24 \pm 77.44)$ | $(46.13 \pm 290.98)$ | $(34.69 \pm 112.84)$ | $(57.14 \pm 141.93)$ | $(1.97 \pm 24.99)$ |
| **MEDIAN-AD** | 0.21 | -0.25 | -0.40 | -0.29 | 0.20 | 0.03 |
| | ( 0.45) | ( -0.07) | ( 0.02) | ( 0.27) | ( 1.12) | ( 0.00) |
| **MEDIAN-RD** | 15.80 | -3.03 | -6.24 | -3.08 | 2.22 | 1.45 |
| | ( 18.01) | ( -1.19) | ( 0.21) | ( 3.01) | ( 9.86) | ( 0.14) |
| **NDATA** | 1279 | 160 | 387 | 141 | 164 | 29488 |
| | (4087) | (467) | (1988) | (893) | (1069) | (92864) |

**Table A1.** Mean and Median Absolute and Relative Differences between CS-UVI Sat/Model Measurements and CS-UVI-RADIO and at ST-DENIS. Mean and Median Absolute and Relative Differences between ALL-SKY-UVI Sat/Model Measurements and ALL-SKY-UVI-RADIO and at ST-DENIS in parentheses

| STATS | Comparison of SAT/MODEL with CS-UVI-RADIO (ALL-SKY-UVI RADIO) at ANTANANARIVO | | | | |
|---|---|---|---|---|---|
| | UVI-CAMS | UVI-OMUVBG | UVI-TROPOMI | UVI-GOME-SN | UVI-TEMIS-SN |
| **MEAN-AD** | $0.14 \pm 0.84$ | $-0.96 \pm 1.16$ | $-1.43 \pm 0.61$ | $-1.10 \pm 1.27$ | $-1.25 \pm 1.11$ |
| | $(0.18 \pm 2.64)$ | $(0.10 \pm 2.24)$ | $(-1.35 \pm 1.91)$ | $(-1.00 \pm 3.23)$ | $(0.77 \pm 3.20)$ |
| **MEAN-RD** | $4.03 \pm 18.53$ | $-12.16 \pm 15.84$ | $-18.94 \pm 5.65$ | $-10.09 \pm 13.80$ | $-11.96 \pm 13.22$ |
| | $(21.40 \pm 113.35)$ | $(13.53 \pm 58.09)$ | $(-8.93 \pm 61.87)$ | $(1.39 \pm 58.78)$ | $(26.44 \pm 139.83)$ |
| **MEDIAN-AD** | -0.01 | -0.86 | -1.38 | -1.02 | -1.41 |
| | ( 0.05) | ( 0.04) | ( -1.26) | ( -1.01) | ( -0.26) |
| **MEDIAN-RD** | -0.55 | -10.98 | -19.11 | -11.11 | -14.13 |
| | ( 2.13) | ( 0.71) | ( -18.02) | ( -10.10) | ( -2.59) |
| **NDATA** | 564 | 92 | 203 | 111 | 142 |
| | (3337) | (427) | (1796) | (816) | (992) |

**Table A2.** Mean and Median Absolute and Relative Differences between CS-UVI Sat/Model Measurements and CS-UVI-RADIO and at ANTANANARIVO. Mean and Median Absolute and Relative Differences between ALL-SKY-UVI Sat/Model Measurements and ALL-SKY-UVI-RADIO and at ANTANANARIVO in parentheses

| | Comparison of SAT/MODEL with CS-UVI-RADIO (ALL-SKY-UVI RADIO) at MAHE | | | | |
|---|---|---|---|---|---|
| STATS | UVI-CAMS | UVI-OMUVBG | UVI-TROPOMI | UVI-GOME-SN | UVI-TEMIS-SN |
| **MEAN-AD** | $-0.82 \pm 1.44$ ($-0.93 \pm 2.94$) | $-3.22 \pm 1.83$ ($-2.16 \pm 2.95$) | $-4.80 \pm 1.60$ ($-3.48 \pm 2.88$) | $-5.75 \pm 2.51$ ($-3.41 \pm 4.29$) | $-4.52 \pm 1.82$ ($-1.22 \pm 4.99$) |
| **MEAN-RD** | $-1.23 \pm 28.57$ ($11.52 \pm 140.58$) | $-26.62 \pm 14.30$ ($-6.28 \pm 101.10$) | $-33.43 \pm 10.43$ ($-24.98 \pm 31.94$) | $-30.65 \pm 14.98$ ($-6.92 \pm 69.82$) | $-24.46 \pm 13.08$ ($23.62 \pm 144.59$) |
| **MEDIAN-AD** | -0.04 ( -0.39) | -3.31 ( -2.63) | -4.88 ( -3.73) | -5.99 ( -4.59) | -4.91 ( -3.19) |
| **MEDIAN-RD** | -8.91 ( -16.41) | -29.49 ( -24.85) | -34.58 ( -32.70) | -32.25 ( -28.55) | -27.11 ( -19.15) |
| **NDATA** | 1133 (3525) | 89 (392) | 253 (2132) | 72 (635) | 94 (881) |

**Table A3.** Mean and Median Absolute and Relative Differences between CS-UVI Sat/Model Measurements and CS-UVI-RADIO and at MAHE. Mean and Median Absolute and Relative Differences between ALL-SKY-UVI Sat/Model Measurements and ALL-SKY-UVI-RADIO and at MAHE in parentheses

| | Comparison of SAT/MODEL with CS-UVI-RADIO (ALL-SKY-UVI RADIO) at ANSE QUITOR | | | | |
|---|---|---|---|---|---|
| STATS | UVI-CAMS | UVI-OMUVBG | UVI-TROPOMI | UVI-GOME-SN | UVI-TEMIS-SN |
| **MEAN-AD** | $0.06 \pm 1.49$ ($-0.20 \pm 1.82$) | $-0.43 \pm 2.36$ ($-0.36 \pm 2.19$) | $-1.28 \pm 1.69$ ($-1.19 \pm 1.80$) | $-1.88 \pm 2.58$ ($-1.42 \pm 2.71$) | $-0.55 \pm 3.34$ ($-0.39 \pm 3.06$) |
| **MEAN-RD** | $12.02 \pm 91.52$ ($6.85 \pm 69.67$) | $17.36 \pm 163.36$ ($12.16 \pm 114.22$) | $-10.93 \pm 26.62$ ($-8.71 \pm 35.26$) | $26.16 \pm 319.83$ ($0.12 \pm 121.81$) | $56.15 \pm 376.30$ ($18.21 \pm 179.43$) |
| **MEDIAN-AD** | -0.06 ( -0.14) | -0.77 ( -0.66) | -1.28 ( -1.25) | -2.23 ( -1.75) | -1.75 ( -1.49) |
| **MEDIAN-RD** | -9.28 ( -9.20) | -12.56 ( -9.45) | -16.65 ( -17.68) | -18.47 ( -17.10) | -14.51 ( -13.31) |
| **NDATA** | 996 (3956) | 204 (516) | 189 (2114) | 95 (752) | 115 (964) |

**Table A4.** Mean and Median Absolute and Relative Differences between CS-UVI Sat/Model Measurements and CS-UVI-RADIO and at ANSE QUITOR. Mean and Median Absolute and Relative Differences between ALL-SKY-UVI Sat/Model Measurements and ALL-SKY-UVI-RADIO and at ANSE QUITOR in parentheses