# Peer review of "UV-Indien Network: Ground-based measurements dedicated to the monitoring of UV radiation over the Western Indian Ocean."

_Earth System Science Data, 2021_

## Author Comment (AC1)

**Authors' Response to Referee #2**

UV-Indien Network ground-based measurements: comparisons with satellite and model estimates of UV radiation over the Western Indian Ocean, **K. Lamy, T. Portafaix, Earth System Science Data, essd-2021-55**
* * *
**Introduction**

We thank the referee for the time and effort he has dedicated to provide us with valuable feedback. We are grateful to the reviewer for their insighful comments and their suggestions.

As mentioned in our response to Referee #1, we have prepared a new version of the manuscript that will be more in line with the objectives and scope of the ESSD and taking into account the comments and suggestions of both referees. This new version of the manuscript is part of our final response to the discussion. (Step 5 of the revision as described here:
https://www.earth-system-science-data.net/peer_review/interactive_review_process.html)

Before we start answering in detail, we would like to point out that we have taken into account the remarks of both referees on the slight discrepancy between the Aims and Scope of ESSD and some sections of the article. We also agree with the referees on the lack of information on site selection, site specificity or instrumental specificities. We are therefore preparing a new version of the manuscript that will include the suggestions and corrections made by the two referees. This new version will follow a structure close to that proposed by Referee #2. Thus, the emphasis of the paper will be shifted from the presentation of the data (Diurnal Cycle of UVR and Clouds) to the description of the instrumented sites, the instrumental characteristics (type of instrument, protocols, calibrations and uncertainties) and the creation of the data set (filtering method, post-processing).

We also hope that the restructuring and shifting of the focus of the paper has helped to realign the manuscript with the ams and scope of ESSD.

Below we provide detailed answers to the main questions raised by the Referee #2. We refer to this part as Part 1: Answer to the General Comments. Our responses to the specific comments are detailled in Part 2 of this document.

**Part 1: Answer to the General Comments**

**General Comments of the Reviewer to the Author:**

A useful manuscript detailing UV measurements in a part of the world that has, until now, had limited coverage. I expect it should find a suitable home somewhere, but as Reviewer #1 notes, it does appear to fall slightly outside the remit of EESD by a strict reading of the journals aims and scope. This could be addressed by a shift in emphasis to providing more details about the ground instruments and their capabilities and calibrations, and a summary of the resulting data, and less emphasis on the intercomparison to satellite datasets. However that specific point I leave for the editor.

Aside from the issue of whether the manuscript properly fits within EESD's remit, I think the manuscript would anyway be improved by this shift in emphasis and some restructuring. There would be much to be gained by adding details of the calibration procedures, more details on the sites and their localities, more details about what instruments are running when at each station, more details about the data processing. This would enable the reader to assess the quality of the resulting data, which is not really possible at present. The comparison to satellite data tells us more about the disparities between different satellite and model products than it does about the ground data — ground data usually being used to validate satellites, rather than the other way round. I found the two sections comparing ground based data to satellite data less helpful, perhaps because the thrust is that there are differences between the Bentham and the satellites due to their different footprints and processing techniques, but little effort to resolve these differences. As a result it tells us little about the Bentham and UVR data. More details and depth on the calibration procedure and instrument characterisation on the other hand would assist here. The conclusions are also heavily biased to interpreting the satellite aspect rather than the novel ground data.

In general I would recommend the authors restructure the paper. The introduction is mostly fine. Then perhaps something as follows:

- site selection (how were they chosen)

- site details (latitude, longitude, altitude, urban, rural, general surrounding habitat, other ancillary measurements e.g. SAOZ)

- instruments and which at which sites

- instrument maintenance protocols and frequencies, characterisations (cosine error, spectral responses), calibrations and data processing (include levels of data)

- data products and pathways (leading on from data processing)

- presentation of summary data: diurnal variation of UVR and CF at each site, monthly means / annual variation, depending on length of measurement period

I think most of this information will be available to the authors if it is not already in the manuscript, but, for example, putting all the information about calibrations together will help (at present it is spread between sections 2.1 and 3.1).
* * *
**Author's Response:** We agree with the reviewer on a necessary restructuring of the manuscript. This has been done and a new version of the manuscript is available. The new manuscript now focuses on the characterisation and choice of instrumented sites, the instruments (characteristics, calibrations, maintenance, data processing). The data presentation part has been lightened but is still present, it contains the comparison with other satellite estimates and UVI models, the ranges of variables of the datasets over the day or over the months of the year. We believe that this presentation of the data (section 4 of the new manuscript) provides the future user with key elements for understanding the ranges of values present in the dataset in order to be able to use the network data appropriately. This part therefore also plays its role of data quality assurance. We also hope that the restructuring and shifting of the focus of the paper has helped to realign the manuscript with the ams and scope of ESSD.

**Part 2: Answer to the Specific Comments**

**Specific points of the Reviewer to the Author:**
* * *
RC: ll 1-2 Rephrase to make it clear that the spectroradiometer is a single instrument shared across the network, rather than each site having a spectroradiometer. Likewise for the all sky cameras that not all sites benefit from such an instrument.

AR: Corrected
* * *
RC: l 2 "Homogenously" is not quite correct, there are not an equal number in each country; perhaps "spatially" or "geographically" would be better. How these sites were selected would also be interesting to include.

AR: Corrected
* * *
RC: l 19 "increases more than 4". I presume this does not mean a factor of 4, but it should be made clearer to the reader.

AR: Corrected
* * *
RC: l 48 Insert comma after "radiation" and remove after "conditions"

AR: Corrected
* * *
RC: l 82 Region should not have a capital here

AR: Corrected because this sentence no longer appears in this form in the new version of the manuscript.
* * *
RC: ll 92-95 This would be better changed to "all sites have operational UVR instruments (...), and four stations (...) have operational all sky cameras" or similar

AR: Corrected. This sentence no longer appears in the new version of the manuscript.
* * *
RC: l 96 Which model spectroradiometer?

AR: Bentham DTMc300. The Bentham model used is now correctly cited in the new version of the manuscript.
* * *
RC: l 97 How are the TOC and calibration measured simultaneously? Or do the authors mean that the calibration is dependent on TOC (and also SZA?) and applied after measurement? How was the calibration obtained? How often is it reassessed?

AR: We mean that the calibration is dependent on TOC and SZA and it is applied after measurements. More details on the calibration process can be found in the newer version of the manuscript in section 3.1.2.
* * *
RC: ll 98-100 What is the calibration frequency so far? This is more important than an anticipated future 2 years cycle for the data presented. How is the cross-calibration carried out? How is the Bentham calibrated and what are the details of its calibration / traceablity? Why was the Saint-Denis calibration method changed from the Davos one to the Bentham-derived one? How does this affect the data? Do the authors mean PMOD when they state "Davos"? If so, this should be stated more clearly and acknowledged.

AR: The UVS-E-T 150124 radiometer (installed in St-Denis) was installed in December 2016. It was recalibrated in Davos (PMOD/WRC) from 1 July to 1 August 2017. It was then compared to the Bentham in St-Denis from April to November 2019. This is not the case for the other instruments. The SUV-E radiometers were all compared to the Bentham in St-Denis before being installed at their respective sites. The other UVS-E-T radiometers (Anse Quitor, Antananarivo and Mahé) should have been repatriated to be recalibrated in Davos or at least in St-Denis with the Bentham, unfortunately logistical and financial constraints have slowed down this recalibration process.

Details have been added in the manuscript on the calibration of the radiometers against the Bentham (section 3.1.2) and on the calibration of the Bentham itself (section 3.1.1).

We did mean PMOD/WRC when we said Davos, clearer details and acknowledgements have been added in the manuscript in order to correctly cite this institute.
* * *
RC: l 101 What do the authors mean by mesh-size (graduations in TOC in the lookup table for calibration)? If so, how does this relate to the cost of the instrument?

AR: Corrected, this sentence was a typo and no longer appears in the new version of the manuscript.
* * *
RC: l 102 How much difference is there between OMI TOC and SAOZ TOC at the stations with both, and how much does this affect the UVR calibration?

AR: We have not experimented with doing the calibration twice with OMI and then SAOZ, comparing the differences on the UVI obtained. However, if we take for example the fn(sza,ozone) function obtained by the PMOD/WRC calibration and if we estimate that the difference between the total ozone value measured by OMI and that of SAOZ at Moufia, is less than 1.3% (Toihir et al., 2018), this would induce a difference in calibrated UVI of less than 1% whatever the solar zenith angle.

References :

Toihir, A. M., Portafaix, T., Sivakumar, V., Bencherif, H., Pazmiño, A., and Bègue, N.: Variability and trend in ozone over the southern tropics and subtropics, Ann. Geophys., 36, 381–404, https://doi.org/10.5194/angeo-36-381-2018, 2018.
* * *
RC: l 105 It would be useful to know what level 0 and level 1 data is as well.

AR: Details have been added on data processing and the notion of levels (although used internally in our data processing chain) no longer appears in the manuscript. (Section 3.1.3)
* * *
RC: l 107 It would be useful to provide at least brief details of the cloud segmentation algorithm and any known issues or bias in addition to the citation to assist the reader.

AR: Details have been added on the cloud segmentation algorithm (section 3.1.1 in the new version of the manuscript).
* * *
RC: l 96-197 Using the abbreviations AD and RD makes reading the following section much harder. Suggest they be removed and spelt out.

AR: As suggested, we have removed the abbreviations AD and RD.
* * *
RC: l 208 Remove space before period.

AR: Corrected
* * *
RC: l 211 BENTHAM should not be fully capitalised here

AR: Corrected
* * *
RC: ll 231-234 This detail should be left for the figure caption.

AR: Corrected
* * *
RC: l 241 Absolute and Relative Differences should not be capitalised

AR: Corrected
* * *
RC: ll 248-249 Yes UVI-RADIO and UVI-BENTHAM should be aligned, but as the authors are presenting the data, and the quality of the data depends on the quality of the calibration, how well this has been transferred between the two instruments would be very interesting to include.

AR: Details have been addded about the calibration of the UVS-E-T located at St-Denis against the Bentham. Theses details are in section 3.1.2.
* * *
RC: l 252 M-RD: these designations make reading the text harder

AR: We understand the reviewer's point about abbreviations. However, we have chosen to keep abbreviations for these terms as they were very present in this section. We feel that writing these terms out in full would also make them more cumbersome to read, so we have changed M-RD and M-AD to Mean-RD and Mean-AD and med-RD and med-AD to Median-RD and Median-AD for more clarity and less clutter.
* * *
RC: l 289 Likely smaller than 100km as well, but including some details of the sites' local environment would complement this point

AR: The environmental conditions have been described in Table 1 and in section 2.2 of the new manuscript.
* * *
RC: l 300 How long is the measurement period? Please provide details.

AR: The measurement period used to make these monthly averages (formerly climatology) is the same as for Figure 5: Anse Quitor (start of measurements (2017) to 06/2020), Antananarivo and St-Denis (start of measurements (2016) to 12/2020) and Mahé (start of measurements (2017 to 11/2020). These details have been added in the manuscript.
* * *
RC: l 317 Enhancement suggests a ratio (or factor), but as before do the authors mean a difference of 2 to 3?

AR: We were talking about an absolute increase in UVI of 2 to 3 units. This is now made explicit in the manuscript.
* * *
RC: l 328 How long is the data set being discussed?

AR: It depends on the station, the periods of the radiometers of each station are now shown in figure 2 of the new version of the article. The length of the data sets discussed here are about 4 years for St-Denis and Antananarivo, about 3.5 years for Anse Quitor and about 3 years for Mahé.
* * *
RC: l 369 Rearrange sentence so does not start with "10 am"

AR: Corrected
* * *
RC: ll 380-390 This information would be better in a "data availability" section

AR: We are not sure we understand this comment because lines 380-390 were in the "Data availability" section.
* * *
RC: Fig 1: It would be useful to distinguish the stations, perhaps by colour, according to the instruments that monitor at each site.

AR: The map has been updated to show the presence or absence of a Bentham spectroradiometer, camera or radiometer.
* * *
RC: Fig 2: This mainly shows satellite timelines. It would be better to show the timelines of the UVR instruments which give most of the geographic coverage and the all sky cameras which are the focus of the manuscript

AR: Both figure are now present in the manuscript. Timeline of UV-Indien instrument are among the main figure. Timeline of satellite, models and Bentham and radiometer at St-Denis are in appendix.
* * *
RC: Fig 5: Is this UVR data or Bentham data?

AR: This is UVR data.
* * *
RC: Fig 6: I am not sure whether the length of the data series is sufficient to call these "climatologies"

AR: As stated in the text (l328 in the old version of the manuscript), we agree with the Reviewer on this point. The measurement periods are not long enough to be considered as climatologies. We have therefore renamed them "monthly averages" in the new version of the manuscript.

---

## Author Comment (AC2)

**Authors' Response to Referee #1**

UV-Indien Network ground-based measurements: comparisons with satellite and model estimates of UV radiation over the Western Indian Ocean, **K. Lamy, T. Portafaix, Earth System Science Data, essd-2021-55**
* * *
**Introduction**

We appreciate the time and effort put into this article by the referee and thank him for his detailed and thorough comments.

Before we start responding in detail, we would like to point out that we have taken into account the remarks of the two referees on the slight discrepancy between the objectives and scope of the ESSD and some sections of the article.

We have therefore prepared a new version of the manuscript with an updated structure that will include the suggestions and corrections proposed by the two referees.

The focus of the paper has now shifted from the presentation of the data (diurnal UV and cloud cycle) to the description of the instrumented sites, the instrumental characteristics (instrument type, protocols, calibrations and uncertainties) and the creation of the dataset (filtering method, post processing). This new version is provided with our final response and all answers to the questions raised during the discussion (step 5 of the revision as described here:
https://www.earth-system-science-data.net/peer_review/interactive_review_process.html).

We hope that the restructuring and shifting of the focus of the paper has helped to realign the manuscript with the ams and scope of ESSD.

Below we provide detailed answers to the main questions raised by the Referee. Our response is structured in two parts. The first part concerns the Referee's general comment. As this general comment is very insightful and detailed, we have decided to divide it into several points. Our responses to these points form the first part of our response.

The second part of our response addresses the specific comments (as distinct from the general comments). We thank the referee for his specific comments that were helpful in improving this manuscript.

**Part 1: Answer to the General Comments of the Referee #1 to the Author**
* * *
*__Referee's Comment 1:__ This article presents a new data set of UV index observations for an area of the world that generally lacks observations. A comparison is presented with UV indices derived from various satellite products. On the whole, these items are worthy of publication in some venue. However, based on the provided review guidelines as I understood them, I could not confirm that the submission meets the objectives of a data description article for the Earth System Science Data journal. Quoting from the aims and scopes of the ESSD (https://www.earth-system-science-data.net/about/aims_and_scope.html),*

*"Articles in the data section may pertain to the planning, instrumentation, and execution of experiments or collection of data. Any interpretation of data is outside the scope of regular articles. Articles on methods describe nontrivial statistical and other methods employed (e.g. to filter, normalize, or convert raw data to primary published data) as well as nontrivial instrumentation or operational methods. Any comparison to other methods is beyond the scope of regular articles."*

*This article does describe how the data was gathered and processed, including calibration information, but this is actually a minor portion of the text and mostly references other sources. The bulk of article describes interpretation of the data, for example the presence of cloud enhancement, and comparisons of radiometer data with results determined by other methods (satellite and model-based products). The short summary states that the article concerns the validation of a new dataset from the measurement of ultraviolet radiation, but the validation described in the lines 110-115 is actually of the satellite and model based numerical products given the ground-based radiometric measurements.*
* * *
**Author's Response 1:** We do not believe that the bulk of the article you refer to (we assume this is part 4 of the article?) can be considered solely as an exercise in data interpretation.

We do not deny that there are elements of explanation of the data, but these explanatory notes are used here to ensure the quality of the data set.

We consider that this allows us to provide key elements for the correct use of the data by the future user of the data. It is therefore about providing information about the data for the purpose of better understanding the instrument and the data but not for the sole purpose of understanding and studying the variability of a physical phenomenon although these two objectives are closely related.

Therefore, in this section we represent the instrumental measurement data and provide a quantitative description of the variability of these data across stations and thus across latitudes, longitudes or environmental situations. These descriptions are made without trying

to explain them. From these representations, we provide short elements to understand the variability of the data set.

The example of the increase in UV by clouds is very telling. This phenomenon, which is very common and can play an important role in the UV measurement, is observed for all measuring stations described in this article. It is therefore necessary to mention it. The article does not investigate the mechanisms and amplitudes associated with the phenomenon of UV enhancement by clouds, but this phenomenon is mentioned and its possible impact on the amplitudes of the observed values is reported. This assures the future user that the maxima observed in the dataset are not the result of poor data quality but the result of an existing physical phenomenon.

In lines 110-115 we only mention the objectives of this network and the possible uses of this network of instruments. Although one of the objectives of the network is to serve as a reference to validate future satellites, models or products. The quality and validity of the measurements of this network must first be ensured against instruments and models already validated and operational before the installation of this network, which is done in the rest of the paper, especially in section 3.2.

However, we agree that more detail is needed on the choice of instrumented sites, the environmental conditions associated with these sites, the characteristics of the instruments and the associated calibration and data processing protocols. As indicated in the introduction, we have therefore restructured the manuscript to focus on these issues. We have chosen to retain some elements of Section 4 because, as explained here earlier, we believe it is important to provide the future user with key information about the variability of the data set.
* * *
> **_RC 2:_** _Moreover, a stated criterium of a ESSD contribution is that it should be easy for any interested person to use the associated archived data, but the data available at https://zenodo.org/record/4572026#.YH8SL2gpC9Y doesn't quite meet that standard. The meta data should provide more information, including what zone is used for time (UTC? LT?), what is a UV index, and what is meant by an "Instant Reading". In one of the data sets, AnseQuitor, the same observation occurs in triplicate at each time point which appears to be an error. The files include data at "night" (SZA >90 but not reported) which are all zero and makes the files unnecessarily long (zero data is omitted for the St-Denis data). The cloud fraction data includes the red-blue ratio the derivation of which is not described in the article. Finally, the data doi only includes the radiometer (Kipp Zonen erythemal) data and not the other data referenced in the paper, including the Bentham spectroradiometer data, TOC and UVI derived from the satellite and model products._
* * *
**AR2:** The data is accessible via 2 databases (zenodo and WOUDC). WOUDC specifies the timezone, which is not the case in the current version on zenodo database. This will be corrected in a version 2 soon available on this site, together with corrections concerning

triple data (Anse Quitor) and unnecessary night measurements (SZA >90). We thank the reviewer for his vigilance.

The ZENODO data d.o.i includes the radiometer (Kipp Zonen erythemal) and cloud fraction; it is not the case for the WOUDC repository. Indeed, only the UV and cloud fraction data are concerned by this article because of their originality.The other data mentioned in the comparisons are freely available on the websites of the organisations concerned. These elements are referenced in our articles so that a user can easily find them. These other data have their own doi (Bentham spectroradiometer, OMI TOC, UVI from satellite or model).
* * *
> **_RC3:_** Calibration procedures are mentioned in the article but details are omitted, in particular when calibrations were performed for the data presented.  The article does mention plans for future operation and procedures for future calibrations.  This information will be of interest to the reader but is not directly relevant to the archived data.  The organization would be improved by segregating such information in a separate section on titled something like "future operations".
>
> ---------------------------------------------------------------------------------------------------

**AR3:** We agree with the referee on this point. The calibration section is now more detailed in the newer version of the manuscript. Nevertheless, in order to answer your questions, more details concerning the calibrations are provided below in the specific commentary associated with the calibration (Part 2 Author's Response 1).
* * *
> **_Referee #1 conclusions before specific comments:_** This is the first paper I have reviewed for ESSD and perhaps I have applied the criteria for publication too narrowly.  For this reason, I have not recommended outright rejection and leave it to the editor to make the final judgement.  Whatever course is taken,  the mss and data sets with revisions should be appropriate for publication in this or some other venue for reporting geophysical results.  In the revision, I recommend attending to the points made above and a number of specific issues as follows:

**Part 2: Answer to the Specific Comments**

> ***Referee's Comment 1:*** *3.1 Calibration – This section refers to results on Radiometer ADs and RDs relative to the Bentham spectroradiometer but only qualitative description is given. Comparing this section and the data section lines 95-109, I am confused about what calibrations were used and how they were done. Line 97 says that the radiometers will be calibrated every two years. How about the calibrations for the presented data? In section 3.1, differences are described between radiometer based and spectroradiometer based UVI for the "recent recalibration". What was the calibration of the radiometers for this comparison? Later in the paragraph, it is stated that "differences were used to recalibrate the radiometer". But isn't that what is usually done in a calibration against a reference instrument?*

**Author's Response 1:** We agree with the Referee that this section may be confusing for the reader. We are currently reworking a new version of this section separated into several parts with more qualitative and quantitative details on the calibration of UVS-E-T radiometers and on the calibration of SUV-E radiometers.

To briefly answer the questions raised by the Referee;:

The calibration of the data presented in the manuscript (Antananarivo, Anse Quitor, Mahe and St-Denis) is the manufacturer's calibration for the first three and the manufacturer's calibration followed by a calibration performed in Davos for the Saint-Denis radiometer.

Recent calibration using an intercomparison with the BENTHAM spectroradiometer measurements was done for the SUV-E Radiometers (installed in Fort Dauphin, Moroni, Diego Suarez and Juan de Nova Island), the BENTHAM is considered as the reference in this calibration process.

We plan to recalibrate all radiometers (SUV-E and UVS-E-T) every 2 to 3 years by re-deploying them for a few months on the same site as the Bentham (Réunion Island, Moufia site). This procedure requires the physical pick-up of the radiometer at each site and the exchange with a recalibrated instrument at La Reunion Island. Indeed, postal services are not efficient in this part of the world (risk of loss or theft) and for some sites we do not have local correspondents allowing a secure shipment. Unfortunately the current health situation has delayed this process since no commercial flights are operated between Reunion, Madagascar, Seychelles, or Comoro Island, since March 2020.
* * *
> ***RC2:*** *What are the criteria for determining which measurement are "clear-sky" and what is meant by a "clear sky day" Please specify. The text says, e.g., that there are 16% "clear sky days" at Antananarivo, but the data set AntananarivoSKYCAMVISION has < 1% of 30s resolution data with CF=0.*

**AR2:** This filtering was done manually for 1h intervals. The filtering process is as follows: each daily UVI and cloud fraction profile is plotted together with an estimate of the clear sky

profile using the Madronich analytical formula. To calculate the Madronich UVI estimate, we use the total ozone column from either the OMTO3d satellite product or a co-located ground-based instrument (the SAOZ for the St-Denis station).

An observer then selects the one-hour windows considered as clear sky according to the following criteria:

- Difference between observed and estimated UVI (according to the Madronich Analytical formula)

- Presence of clouds, a cloud fraction threshold of 0.25 is set. A UVI measurement performed

- Shape and regularity of the UVI curve during the day. A Gaussian (or semi-Gaussian) curve indicates a day (or half-day) not affected by the presence of clouds. As clouds generally have a high temporal variability, rapid development of clouds usually results in rapid variations of UVI over a few minutes.

- Cloud cover can also be quasi-homogeneous and quasi-constant over the day, which does not lead to sudden variations in UVI, and this case is also excluded from the filtering.

References:

Madronich, S. (2007), Analytic Formula for the Clear-sky UV Index. Photochemistry and Photobiology, 83: 1537-1538. https://doi.org/10.1111/j.1751-1097.2007.00200.x

These details have been added to the manuscript.
* * *
**_RC3:_** _Line 93 "on station Juan de Nova" Should read "one station at Juan de Nova"?_

**AR3:** Corrected. This sentence no longer appears in the new version of the manuscript.
* * *
**_RC4:_** _Line 95 "All stations are now equipped with a Kipp & Zonen UVS-E-T broadband radiometer."_

_Table 3 and later text states that some of the radiometers are the SUV-E model_

**AR4:** Corrected. This sentence no longer appears in the new version of the manuscript.
* * *
**_RC5:_** _Line 96 "The raw UV measurements obtained by the radiometers are reprocessed considering the calibrations and TOC measured simultaneously."_

_Please describe and/or give reference for the reprocessing procedure. The reference given in Table 2 describes how the KZ radiometer was calibrated in Davos, but what is the current procedure?_

**AR5:** See AR1
* * *
**RC6:** *Line 101  "on a smaller mesh size"*

*Not clear what this means*

**AR6:** Corrected, this sentence was a typo and no longer appears in the new version of the manuscript.
* * *
**RC7:** *Line 212 "Table 4 presents the different radiometers and their current locations, along with the date of the next calibration"*

*This information is in Table 3*

**AR7:** Corrected.
* * *
**RC8:** *Line 225 Figure 2 shows the period covered by each data set at these four stations.*

*The figure only has one line covering data for the broad-band radiometers, does not show each station separately*

**AR8:** Each station are now represented sepately in the new version of the manuscript. Timeline of every UV-Indien instrument are among the main figure (Figure 2). Timeline of satellite, models and Bentham and radiometer at St-Denis are in appendix.
* * *
**RC9:** *Line 315 – UV enhancement by cloud scattering is a well known and widely occuring phenomenom, would be appropriate to cite the Sabburg and Wong (2000) paper here, also see Badosa et al. (2014)*

**AR9:** Corrected.
* * *
**RC10:** *Line 347 "The density of the corresponding data set for each month of the year is represented in Figure 7b"*

*Please define what is density in this context, proportion of what?*

**AR10:** In this context, density is the ratio of the number of measurements for each month to the total number of measurements normalized between 0 and 1. This information is now included in the manuscript.
* * *
*__RC11:__ Line 394 - The correlation coefficient between the satellite or model estimates and the ground-based measurements was greater than 0.9 at all stations except Maheİ   and for all datasets except OMUVBG.*

*Revise to state that correlation applies only for clear sky conditions*

__AR11:__ Corrected in the new version of the manuscript.
* * *
*__RC12:__* 6-Conclusions

*Perhaps the authors can comment on why of the three stations at similar south latitude, Antananarivo, Anse Quitor,  and St-Denis,  the UV-index, even for clear sky, seems to be systematically lower at St-Denis.  For example, the noon mean CS UVI is ~10 at St.-Denis but ~12 at Anse Quitor.  The maximum CS UVI is ~18 at Anse Quitor and ~17 at Antananarivo, but <15 at St-Denis.*

__AR12:__ Although the stations of Antananarivo, Anse Quitor and Saint Denis are almost at the same latitude, the environmental conditions are very different. Total ozone variations do not exceed 10 DU between these 3 sites but other conditions influencing UV can vary considerably.

The Antananarivo station is located at more than 1370m a.s.l while Saint Denis and Anse Quitor are at 82m and 32m respectively. The maximum clear-sky UV index is expected to be higher at Antananarivo than at Anse Quitor but the UVI is affected by the very high presence of air pollution at Antananarivo.

The differences between Saint-Denis and Anse Quitor are also due to air quality. The agglomeration of Saint-Denis has about 200k inhabitants, the traffic is important and although this coastal city is under the trade winds, the presence of a thin aerosol layer can be measured all day long. Anse Quitor is part of the Plaine Corail district, the population of this district is about 3000 and there is very little traffic.

These elements are added to the descriptive part 2.2 of the new manuscript which includes details on the choice of sites, on the environmental conditions or on the available ancillary data for each station (AOD, total ozone, etc.), in accordance with the recommendations of reviewer #2
* * *
*__RC13__ Figure 1,  Map – add Latitude and Longitude*

__AR13:__ The map has been updated to show the presence or absence of a Bentham radiometer, camera or spectrometer.
* * *
*RC14: Figure 3 caption– What do the gray bars represent in the histogram sub figures? This also applies to the Appendix figures*

**AR14:** The grey bars represent the distribution of points by intervals of UVI values (for clear sky data only). The curves (blue and red) represent estimates of the probability density functions (for clear-sky and all-sky data respectively).
* * *
*RC15: Figure 5. The caption reads "Diurnal Cycle of UVI at ST-DENIS."*

*The figure actually has all stations, not just ST-DENIS.*

**AR15:** Corrected
* * *
*RC16: Figure 7 & 8 captions – Describe the data density subplots (b) and (d)*

*Figure 7c and 8c – The annual mean difference CS-AS UVI seems to follow very closely the DJF mean – Is this correct? Seems like the mean should be approximately in the middle of all the monthly means, but this is not the case for much of the plot, especially in 8c.*

**AR16:** There was an error in the legend and subtitle of figures 7 and 8. These are not averages but maxima.

In this figure, the difference in UVI maxima in clear and cloudy sky for the whole dataset (, and for the seasons (DJF, JJA, MAM and SON) is shown.

The fact that the DJF curve is quasi-confluent with the ALL curve for an instant t of the day (formerly 'mean' curve, the previous name was a typo) indicates that the differences in UVI maxima observed over the whole year at this instant t of the day are from December, January and February.

In other words, if we write in subscript the season associated with the UVI maximum at a time t1 and the caracteristic (CS: clear-sky or AS: all-sky), and let's take for example the JJA season, we can have the following cases::

1) The JJA seasonal curve is the same as the curve for the whole period (ALL), i.e:

$$\max(UVI_{JJA,CS})(t_1) - \max(UVI_{JJA,AS})(t_1) == \max(UVI_{ALL,CS})(t_1) - \max(UVI_{ALL,AS})(t_1)$$

In this case, the maxima observed at time t1 in CS and AS for the whole period are the same as those observed for the JJA season.

2) The JJA seasonal curve is above or below the curve for the whole period, i.e:

$$\max(UVI_{JJA,CS})(t_1) - \max(UVI_{JJA,AS})(t_1) > \max(UVI_{ALL,CS})(t_1) - \max(UVI_{ALL,AS})(t_1)$$

or

$$\max(UVI_{JJA,CS})(t_1) - \max(UVI_{JJA,AS})(t_1) < \max(UVI_{ALL,CS})(t_1) - \max(UVI_{ALL,AS})(t_1)$$

In this case, $\max(UVI_{ALL,AS})(t_1)$ or $\max(UVI_{ALL,CS})(t_1)$ are from another season.

Indeed, over the whole period, at $t_1$ $\max(UVI_{ALL,CS})(t_1)$ could be observed in December while $\max(UVI_{ALL,AS})(t_1)$ would be observed in June and the resulting difference could then be lower or higher than the difference of the two UVIs (CS and AS) from the same season (here JJA).

Typographical errors have been corrected on the figure and in the text.
* * *
**_RC17:_** _Table 1 – Region location for Juan de Nova station should be Ile Juan de Nova_

**AR17:** Corrected in the new version of the manuscript.
* * *
**_RC18:_** _Reference missing bibliographic information: Pastel, M., Pommereau, J.-P., Goutail, F., Richter, A., Pazmino, A., Ionov, D. V., and Portafaix, T.: Construction of merged satellite total O3 and NO2 time series in the tropics for trend studies and evaluation by comparison to NDACC SAOZ measurements, 2014._

**AR18:** Corrected in the new version of the manuscript.
* * *
**_RC19:_** _References:_

_Badosa, J., Calbó, J., Mckenzie, R., Liley, B., González, J.â  A., Forgan, B. and Long, C.N. (2014), Two Methods for Retrieving UV Index for All Cloud Conditions from Sky Imager Products or Total SW Radiation Measurements. Photochem Photobiol, 90: 941-951. https://doi.org/10.1111/php.12272_

**AR19:** Added in the new version of the manuscript.

---

## Author Comment (AC3)

**Authors' Response to Reviews of**

**UV-Indien Network ground-based measurements: comparisons with satellite and model estimates of UV radiation over the Western Indian Ocean.**

K. Lamy, T. Portafaix
*Earth System Science Data,* `essd-2021-55`
* * *
EC: *Editors' Comment*,     AR: Authors' Response

**EC:** ***Editor's Comments to the Author:***
***Dear authors,***

Thank you for your submission to ESSD. Based on the first two reviews, I recommend that you carefully consider whether or not you can demonstrate the usefulness of this dataset. This purpose of ESSD is to publish data sets that can be used by others. As pointed out by reviewer 2, ground-based observations are usually used to validate satellite data/model output. Although a thorough inter-comparison is presented in the paper, it is not clear which dataset is being validated. Can an argument be made that the UV-Indien network can be considered as the ground-truth and perhaps be used to evaluate models and satellite observations? If this can not be demonstrated, then I would agree with the reviewers that it is not suitable for publication in ESSD and advise you to consider publication in another journal.

On the other hand, if a strong argument for this can be made, then I suggest the authors follow the recommendations of both reviewers and shift the focus of the manuscript to be more on the details of how the data was gathered and processed as outlined by the reviewers. In addition, re-orient the intercomparison of the data products to demonstrate how the UV-Indien network can be used to validate other data products (e.g. satellite observations, model output). Efforts should also be made in making the data product more user friendly.

If you choose to submit a revised version, please make sure to address each of the reviewers' comments, point by point.

Please let me know if you have any further questions.

Best regards,

Nellie Elguindi

AR:

The dataset from the Indian UV network is of high quality, it is composed of regularly recalibrated mid to high range instruments, the data processing procedure before distribution is thorough and the feedback from reviewers on the files has been corrected. We therefore believe that the Indian UV network can be considered as the reference for ground-based measurements and be used to evaluate and validate models. It is also very useful for other users in a particularly data poor region. This region also shows a diversity of environmental observation sites sampled as best as possible by the UV-India network. These arguments are outlined in the new version of the manuscript that we have produced in accordance with the recommendations of the two reviewers and in response to all their concerns. We believe that the focus of the manuscript is now well aligned with the Aims and Scope of ESSD.

Best regards,

Kevin Lamy

---

## Author Response (AR2)

**Authors' Response to Reviews of**

**UV-Indien Network: Ground-based measurements dedicated to the monitoring of UV radiation over the Western Indian Ocean.**

K. Lamy, T. Portafaix, C. Brogniez
*Earth System Science Data,* `essd-2021-55`
* * *
**EC:** *Editor's Comment*,    **RC:** *Reviewers' Comment*,    AR: Authors' Response

**EC:** Topical Editor's Comments to the Author:

> **Dear Dr. Lamy,**
> **Thank you for submitting your revised manuscript. Referee 1 has some additional comments (see below) which need to be addressed point by point and submitted along with the revised manuscript.**
> **Please let me know if you have any questions.**
> **Best regards,**
> **Nellie Elguindi**

AR: Author's Response

Dear Editor, Thank you for these new corrections. Please find below our point-by-point responses to Referee 1's comments. Best regards,

K. Lamy, T. Portafaix, C. Brogniez

**RC:** Referee 1 General Comments:

> This revised version addresses the concerns about the suitability of the paper for the journal expressed in the first set of reviews. This has improved its presentation as a data description article for the Earth System Science Data journal. In particular, the shift in focus is evident in the change of title (mention of satellites has been dropped) and the detailed information provided on the sites, instrumentation and calibration procedures. Generally, I find the submission more in line with the objectives of a ESSD data description paper. However, I continue to have some concerns about two of the main elements that were mentioned in the review of the first version, namely (1) the extent to which the data set, per se, is the focus of the presentation and (2) having a transparent account of which calibrations were applied over the full period of the publically available data set.

AR: Author's Response

First of all, we would like to thank the referee for the time he spent on our manuscript and the relevant comments that helped to significantly improve the quality of the manuscript. Regarding the last two points of concern of the referee, we have modified the manuscript accordingly. Section 4 has been modified in accordance with the referee's suggestions to restructure the order of the sections, to condense the intercomparison section, and to shift the focus of the intercomparison from the Bentham to the radiometer. Regarding

the second point, calibration, the presentation figure of the instrument and calibration timelines has been improved and information has been added in the manuscript about the Bentham calibration, possible offsets and the future of the dataset. Please find below our detailed answers to these two points.

**RC:**  Referee 1: First Point

> **On the first point, I appreciate the author's view that describing some characteristics of the data, particularly the presence of cloud enhancement, will assist users in proper use of the data. In recognition of this important consideration of the data, the sections on diurnal variation and cloud fraction should be the first parts of section 4. Like the first version, this revised version still leads off section 4 with a lengthy section about the derivation of the satellite/model and inter-comparison with the data. Within the three sections, numbers 3-5, the exposition of the satellite data set and its comparison to the Bentham and radiometers occupies 40% of the text (counting the 15 lines of the conclusion which basically just restate the results in section 4.1). This seems out of proportion to its importance in guiding use of the data.**
>
> **In a paper about a data set the primary focus should be its characteristics, per se, thus I recommend putting sections 4.2 and 4.3 before 4.1. Indeed, having the characteristics of cloudiness described first assists in explaining some of the discrepancies with the satellite/model results. The satellite/model comparison section should be condensed. The comparison should focus on the radiometer results, since that is what is in the published data set. I don't see what is shown in Figures 3 and 4 that is not also shown in the appendix figure A1 and A2. The latter are more relevant, since the Bentham data shown in Figures 3 and 4 are not included in the published data set.**

AR:

As suggested by the reviewer, sections 4.2 and 4.3 dealing respectively with the diurnal variation of the cloud fraction and UVI and the seasonal variability of UVI extremes have been moved upstream from the comparison with other UVI measurements. Sections 4.2 and 4.3 are now section 4.1 and 4.2 respectively.

The former section 4.1 (Comparison with other satellite measurements or models) is now section 4.3. Following the reviewer's suggestion, this section has also been condensed while keeping the essential references and information of the different satellite products and models (spatial and temporal resolution, parameters used). We have chosen to keep this information as it allows us to understand the discrepancies in the UV-Indian dataset and the other datasets. This information also makes it possible to appreciate the fact that the ground-based UV-Indien data, by virtue of their measurement method, take into account all the physical parameters influencing the UVI on the ground, unlike certain models or instruments on board satellites. These differences allow us to give indications of how the dataset can be used both for UVI studies and for satellite or model improvements.

The detailed comparison in this section is now between the St-Denis radiometer and the satellites and models (and no longer between the Bentham and the satellites and models). The corresponding figures and tables have therefore been inverted to reflect this change in the text. (Table 4 <-> Table A1, Figure 3/4 <-> Figure A2/A3)

**RC:**  Referee 1: Second point

> **On calibration, I appreciate the detailed explanation of each of the three types of calibration. But only two of these calibrations are mentioned in Table 3 showing the caibrations by instru-**

> ment: manufacturer's and Bentham. The author's response to reviewers states that in addition to the present Bentham calibration, the manufacturer's and Davos calibration were also used for the Saint-Denis radiometer.
>
> A more complete accounting of calibrations covering the whole published record is needed. This could be conveyed in the paper by using different color codes in Figure 2 for periods covered by different procedures. More importantly, there are some inconsistencies between calibration procedures in terms of how corrections are applied as a function of SZA and ozone (no ozone adjustment for Bentham calibrations [?]). Have the authors examined what are the possible effects of these different calibration procedures on the record? Are there offsets that have been introduced in the record when calibration/correction procedures are changed? Lines 240-245 describe differences in the corrected UVI values between the Bentham calibration and manufacturer's/Davos calibrations. What (if anything) was done to the record to adjust for these differences in the UVI record?
>
> Finally, it appears as though some of the data being reported is from instruments that, for various reasons, were overdue for recalibration at the time the observations were made but are scheduled for new calibrations (according to Table3). Should data users expect a possible round of corrections to these data sets when the "end cals" for a deployment period are known?

AR:   Author's Response

In order to better detail the instrument calibrations and data calibration periods, Figure 2 has been updated with a colour code.The corresponding figure can be found below this answer.

Indeed, the Bentham calibration does not directly include the total ozone column. The Bentham calibration coefficients are functions of the SZA and were calculated from the manufacturer's recalibrated radiometer measurements. These manufacturer's recalibrated radiometer measurements have a calibration coefficient that is a function of the SZA and the total ozone column. Unfortunately, for the Bentham calibration to take the total ozone column directly into account, the radiometers would have to be installed for at least one year at the Moufia site in order to capture all possible ozone and SZA values over the course of a year. The intermediate use of a radiative transfer model could also be used to extrapolate to larger total ozone column values. The impact of taking into account the total ozone column on the Bentham calibration has not yet been studied for these instruments. No offset has been particularly observed between two calibration periods, however we have not yet done any detailed analysis on this subject. We thank the reviewer for raising these points of interest regarding the calibration and thus the quality of the data set. We plan to focus our next study on these aspects. The discussion detailed here is now present in the manuscript in order to inform the user about possible offsets due to calibration changes.

Another round of correction is indeed expected, especially for radiometers whose calibration date has passed. When calibrations are made possible again by the opening of the borders, instruments that have been in place for more than 2 years will be replaced by instruments recalibrated from the Bentham. The replaced instruments will then be recalibrated at the Moufia site before possible redeployment. During this re-calibration, and if it is noted that there is too much drift compared to the previous calibration, it is possible that part of the data set will have to be corrected over a short period and made available to users with an alert. This new correction will also allow a more detailed understanding of possible calibration offsets or the impact of the total ozone column for the Bentham calibration. The available dataset will therefore get a new version and the documentation will also detail the new modifications.

[Figure]

Figure 1: Timeline of the instruments of the UV-Indien network.
The measurement periods of the cameras are shown in purple. The measurement period of the BEN-THAM spectroradiometer is shown in orange. The measurement periods of the radiometers that are either manufacturer-calibrated, Davos-calibrated or BENTHAM-calibrated are shown in green, red and blue respectively.